# Benchmarking of cell type deconvolution pipelines for transcriptomics data

Francisco Avila Cobos [1,2,3✉], José Alquicira-Hernandez [3,4], Joseph E. Powell [3,4,5], Pieter Mestdagh [1,2,5] & Katleen De Preter [1,2,5✉]

Many computational methods have been developed to infer cell type proportions from bulk transcriptomics data. However, an evaluation of the impact of data transformation, pre-processing, marker selection, cell type composition and choice of methodology on the deconvolution results is still lacking. Using five single-cell RNA-sequencing (scRNA-seq) datasets, we generate pseudo-bulk mixtures to evaluate the combined impact of these factors. Both bulk deconvolution methodologies and those that use scRNA-seq data as reference perform best when applied to data in linear scale and the choice of normalization has a dramatic impact on some, but not all methods. Overall, methods that use scRNA-seq data have comparable performance to the best performing bulk methods whereas semi-supervised approaches show higher error values. Moreover, failure to include cell types in the reference that are present in a mixture leads to substantially worse results, regardless of the previous choices. Altogether, we evaluate the combined impact of factors affecting the deconvolution task across different datasets and propose general guidelines to maximize its performance.

[1] Center for Medical Genetics Ghent, Department of Biomolecular Medicine, Ghent University, Ghent, Belgium. [2] Cancer Research Institute Ghent (CRIG), Ghent, Belgium. [3] Garvan Weizmann Centre for Cellular Genomics, Garvan Institute of Medical Research, Sydney, NSW, Australia. [4] Institute for Molecular Bioscience, University of Queensland, Brisbane, QLD, Australia. [5]These authors contributed equally: Joseph E. Powell, Pieter Mestdagh, Katleen De Preter. ✉email: Francisco.AvilaCobos@UGent.be; Katleen.DePreter@UGent.be

Since bulk samples of heterogeneous mixtures only represent averaged expression levels (rather than individual measures for each gene across different cell types present in such mixture), many relevant analyses such as differential gene expression are typically confounded by differences in cell type proportions. Moreover, understanding differences in cell type composition in diseases, such as cancer will enable researchers to identify discrete cell populations, such as specific cell types that could be targeted therapeutically. For instance, active research on the role of infiltrating lymphocytes and other immune cells in the tumor microenvironment is currently ongoing[1–3] (e.g., in the context of immunotherapy) and it has already shown that accounting for the tumor heterogeneity resulted in more sensitive survival analyses and more accurate tumor subtype predictions[4]. For these reasons, many methodologies to infer proportions of individual cell types from bulk transcriptomics data have been developed during the last two decades[5], along with new methods that use single-cell RNA-sequencing (scRNA-seq) data to infer cell proportions in bulk RNA-sequenced samples. Collectively we term these approaches cell deconvolution methods.

Several studies have addressed different factors affecting the deconvolution results but only focused on one or two individual aspects at a time. For instance, Zhong and Liu[6] showed that applying the logarithmic transformation to microarray data led to a consistent under-estimation of cell-type specific expression profiles. Hoffmann et al.[7] showed that four different normalization strategies had an impact on the estimation of cell type proportions from microarray data and Newman et al.[8] highlighted the importance of accounting for differences in normalization procedures when comparing the results from CIBERSORT[9] and TIMER[10]. Furthermore, Vallania et al.[11] observed highly concordant results across different deconvolution methods in both blood and tissue samples, suggesting that the reference matrix was more important than the methodology being used.

Sturm et al.[12] already investigated scenarios where reported cell type proportions were higher than expected (spillover effect) or different from zero when a cell type was not present in a mixture (background prediction), possibly caused by related cell types sharing similar signatures or marker genes not being sufficiently cell-type specific. Moreover, they provided a guideline for method selection depending on which cell type of interest needs to be deconvolved. However, each method evaluated in Sturm et al. was accompanied by its own reference signature for the different immune cell types, implying that differences may be marker-dependent and not method-dependent. Moreover, they did not evaluate the effect of data transformation and normalization in these analyses and only focused on immune cell types.

Here we provide a comprehensive and quantitative evaluation of the combined impact of data transformation, scaling/normalization, marker selection, cell type composition and choice of methodology on the deconvolution results. We evaluate the performance of 20 deconvolution methods aimed at computing cell type proportions, including five recently developed methods that use scRNA-seq data as reference. The performance is assessed by means of Pearson correlation and root-mean-square error (RMSE) values between the cell type proportions computed by the different deconvolution methods ($P_C$; computed proportions; Fig. 1) and known compositions ($P_E$; expected proportions) of a thousand pseudo-bulk mixtures from each of five different scRNA-seq datasets (three from human pancreas; one from human kidney and one from human peripheral blood mononuclear cells (PBMCs)). Furthermore, to evaluate the robustness of our conclusions, different number of cells (cell pool sizes) are used to build the pseudo-bulk mixtures. We observe that the most relevant factors affecting the deconvolution results are: (i) the

data transformation, with linear transformation outperforming the others, (ii) the reference matrix, which should include all cell types being part of the mixtures, iii) a sensible marker selection strategy for bulk deconvolution methods.

## Results

**Memory and time requirements**. While simple logarithmic (log) and square-root (sqrt) data transformations were performed almost instantaneously in R (between 1 and 5 s; see Table 1 for information about the number of cells subject to transformation in each scRNA-seq dataset), the variance stabilization transformation (VST) performed using DESeq2[13] applied to the scRNA-sequencing datasets had high memory requirements and took several minutes to complete (time increasing linearly with respect to the number of cells) (Supplementary Fig. 3). Importantly, DESeq2 v1.26.0 (or above) reduced the running time from quadratic (Supplementary Fig. 27 from Soneson et al.[14]) to linear with respect of the number of cells.

We further evaluated the impact of different scaling and normalization strategies as well as the choice of the deconvolution method. Although the different scaling/normalization strategies consistently have similar memory requirements, SCTransform[15] and scran[16] (two scRNA-seq specific normalization methods; the former uses regularized negative binomial regression for normalization (RNBR)) required up to seven minutes to complete, a 14 fold difference with the other methods, which finished under 30 s (Supplementary Fig. 4).

The bulk deconvolution methods DSA[17], ssFrobenius and ssKL[18] (all implemented as part of the CellMix[19] R package) had the highest RAM memory requirements, followed by DeconRNASeq[20]. Not surprisingly, the ordinary least squares (OLS[21]) and non-negative least squares (nnls[22]) were the fastest, as they have the simplest optimization problem to solve. Regarding the methods that use scRNA-seq data as reference, Dampened Weighted Least Squares (DWLS[23]), which includes an internal marker selection step, resulted in the longest time consumption (6–12 h to complete) whereas MuSiC[24] and SCDC[25] finished in 5–10 mins. Running time and memory usage for the different deconvolution methods is summarized in Supplementary Fig. 5.

**Impact of data transformation on deconvolution results**. We investigated the overall performance of each individual deconvolution method across four different data transformations and all normalization strategies (Fig. 2; Supplementary Fig. 6–7). Maintaining the data in linear scale (*linear* transformation, in gray) consistently showed the best results (lowest RMSE values) whereas the logarithmic (in orange) and VST (in green; which also performs an internal complex logarithmic transformation) scale led to a poorer performance, with two to four-fold higher median RMSE values. For a detailed explanation concerning several bulk deconvolution methods and those using scRNA-seq data as reference that could only be applied with a specific data transformation or dataset, please see Supplementary Methods.

With the exception of EPIC[26], DeconRNASeq[20], and DSA[17], the choice of normalization strategy does not have a substantial impact on the deconvolution results (evidenced by narrow boxplots). These conclusions also hold when repeating the analysis with different pseudo-bulk pool sizes in all datasets tested (collapsing all scaling/normalization strategies and all bulk deconvolution methods (Supplementary Fig. 8) or those using scRNA-seq data as reference (Supplementary Fig. 9)). For these reasons, all downstream analyses were performed on data in linear scale. In terms of performance, the five best bulk deconvolution methods (OLS, nnls, RLR, FARDEEP, and CIBERSORT) and the three best methods that use scRNA-seq

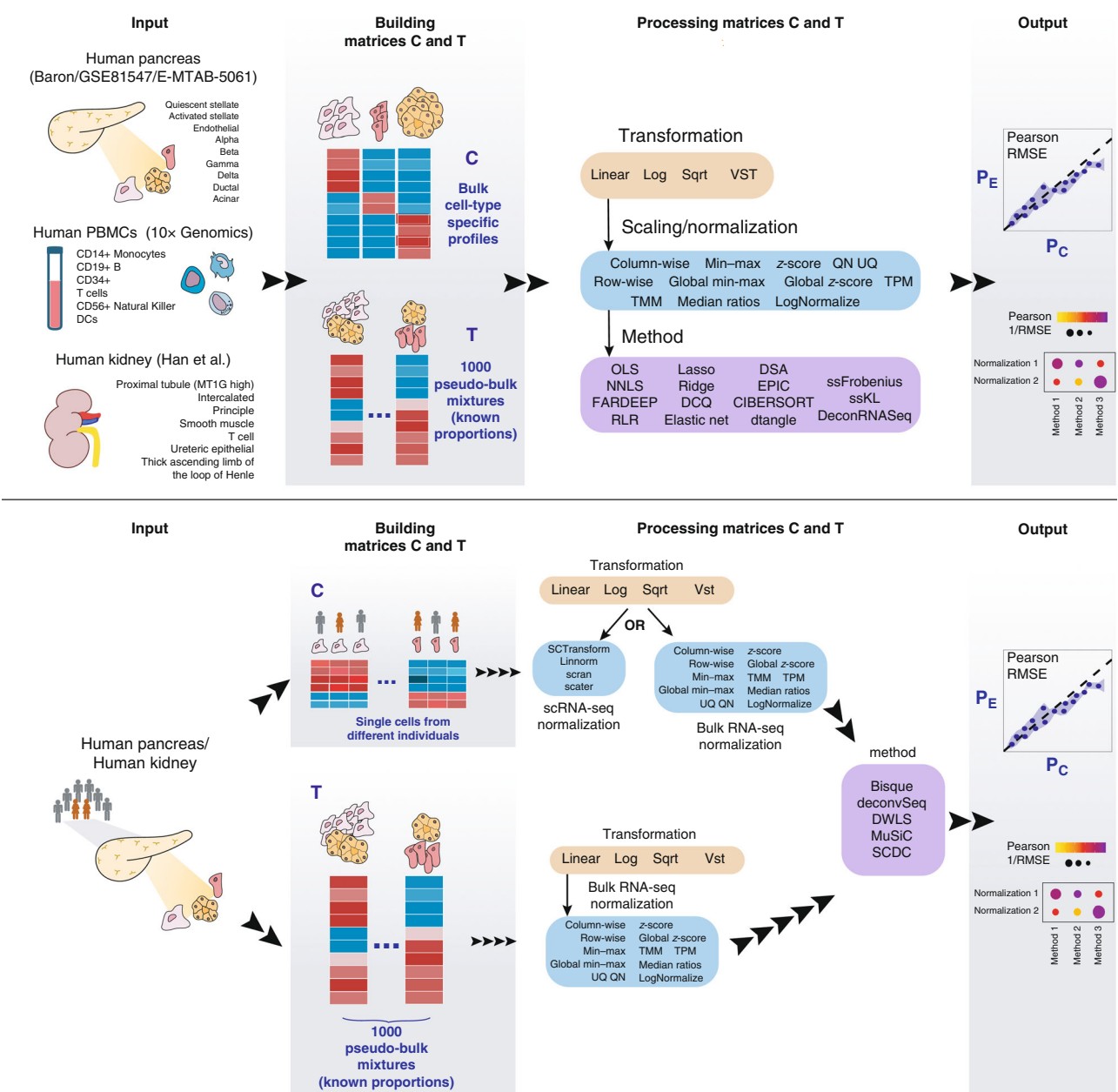

**Fig. 1 Schematic representation of the benchmarking study.** Top panel: workflow for bulk deconvolution methods. Bottom panel: workflow for deconvolution methods using scRNA-seq data as reference. In both cases the deconvolution performance is assessed by means of Pearson correlation and root-mean-square error (RMSE). PBMCs: peripheral blood mononuclear cells, log: logarithmic, sqrt: square-root, VST: variance stabilization transformation, PE: expected proportions, Pc: computed proportions.

data as reference (DWLS, MuSiC, SCDC) achieved median RMSE values lower than 0.05. Penalized regression approaches, including lasso, ridge, elastic net regression, and DCQ performed slightly worse than the ones described above (median RMSE ~ 0.1).

**Different combinations of normalization and deconvolution methods**. It is clear that different combinations of normalizations and methodologies lead to substantial differences in performance (Fig. 2 and Supplementary Fig. 6). Focusing on the data in linear scale, we delved into the specific method and normalization combinations evaluated. Among the bulk deconvolution methods, least-squares (OLS, nnls), support-vector (CIBERSORT) and robust regression approaches (RLR/FARDEEP) gave the best

results across different datasets and pseudo-bulk cell pool sizes (median RMSE values < 0.05; Fig. 3a and Supplementary Figs. 10, 12). Regarding the choice of normalization/scaling strategy, column min–max and column z-score consistently led to the worst performance. In all other situations, the choice of normalization/scaling strategy had minor impact on the deconvolution results for these methods. When considering the estimation error relative to the magnitude of the expected cell type proportions, smaller proportions consistently showed higher relative errors (see Supplementary Figs. 20–23). Of note, quantile normalization always resulted in sub-optimal results in any of the tested bulk deconvolution methods (Fig. 3a, b).

As stated in its original publication, EPIC assumes transcripts per million (TPM) normalized expression values as input. We indeed observed that the choice of scaling/normalization has a big

**Table 1 Details of the five datasets used.**

| Dataset | Biological sample type | Sequencing protocol | Number of individual samples | Number of cell types | Number of cells after QC | Number of genomic features after QC | Median total counts per cell after QC | Median number of non-zero features per cell after QC | Ref |
|---|---|---|---|---|---|---|---|---|---|
| Baron (GSE84133) | Human pancreatic islands | inDrop platform + CEL-Seq protocol | 4 (2 male, 2 female) | 10 | 7692 | 8386 | 4856 | 1723 | 52 |
| E-MTAB-5061 | Human pancreatic tissue and islets | FACS sorting into 384-well plates + Smart-Seq2 | 6 (5 male, 1 female) | 6 | 908 | 13,899 | 329,217 | 5521 | 53 |
| GSE81547 | Human pancreatic tissue | FACS sorting into 96-well plates + Smart-Seq2 | 8 (6 male, 2 female) | 5 | 2068 | 11,694 | 481,825 | 3072 | 54 |
| PBMCs* | Human fresh peripheral blood mononuclear cells | Chromium GemCode Single-Cell Instrument + GemCode Single-Cell 3' Gel Bead and Library Kit (10× Genomics) | 1 | 6* | 10,000* | 2175 | 1142 | 401 | 55 |
| kidney.HCL **(HCL = the human cell landscape) | Human adult kidney (adjacent normal and left kidney) from Han et al. | Microwell-Seq | 2 (males) | 8 | 6393 | 2288 | 557 | 345 | 56,57 |

(*) Since this dataset originally contained six closely related T-cell subtypes (and other people have failed in their attempts of distinguishing them[37,58]) we re-labeled all cells from these sub-types as T-cells. Moreover, to reduce the memory and time requirements needed to run all combinations of data transformation, normalization and methodology, we randomly selected 10,000 cells out of the original 68,000; (**) there were three adult kidney samples available with five cell types in common but, in order to maximize the number of common cell types, we selected Adult 2 and Adult 4 (eight cell types in common; see Fig. 1).

impact on the performance of EPIC, with TPM giving the best results. The semi-supervised approaches ssKL and ssFrobenius (using only sets of marker genes, in contrast to the supervised counterparts which use a reference matrix with expression values for the markers) showed the poorest performances with the highest root-mean-square errors and lower Pearson correlation values (Fig. 3a and Supplementary Fig. 10).

For deconvolution methods using scRNA-seq data as reference (Fig. 3c and Supplementary Fig. 11), we evaluated each combination of normalization strategies for both the pseudo-bulk mixtures (*scalingT*, *y*-axis) and the single-cell expression matrices (*scalingC*, *x*-axis). DWLS, MuSiC and SCDC consistently showed the highest performance (comparable to the top-performers from the bulk methods, see also Fig. 2) across the different choices of normalization strategy (with the exception of row-normalization, column min-max, and TPM). While these results are consistent for deconvSeq, MuSiC, DWLS, and SDCD regardless of the dataset and pseudo-bulk cell pool size, we observed a substantial performance improvement in BisqueRNA when the pool size increased or when the dataset contained scRNA-seq from more individuals (E-MTAB-5061 and GSE81547, with $n = 6$ and 8, respectively) (Supplementary Figs. 7, 11). Note that it was not feasible to evaluate all combinations (empty locations in the grid), see "Incompatible data transformations or normalizations with several deconvolution methods" (Supplementary Notes) for a detailed explanation.

**Impact of the markers used in bulk deconvolution methods**. Based on the previous results, we wanted to evaluate whether different marker selection strategies had an impact on the deconvolution results starting from bulk expression data in linear scale. To that end, we assessed the impact of eight different marker selection strategies (see "Methods") on the deconvolution results using bulk deconvolution methods (Fig. 4 and Supplementary Fig. 13). This analysis was not done for the methods that use scRNA-seq data as reference because they do not require marker genes to be known prior to performing the deconvolution.

The use of all possible markers (*all* strategy) showed the best performance overall, followed by positive fold-change markers (*pos_fc*; negative fold-change markers are those with small expression values in the cell type of interest and high values in all the others) or those on the top 50% of average expression values (*top_50p_AveExpr*) or log fold-changes (*top_50p_logFC*). As expected, the use of random sets of 5 markers per cell type (*random5*; negative control in our setting) was consistently the worst choice across all datasets regardless of the deconvolution method. Using the bottom 50% of the markers per cell type based on average expression levels (*bottom_50p_AveExpr*) or log fold changes (*bottom_50p_logFC*) also led to sub-optimal results. Specifically in the Baron and PBMC datasets, the use of the top 2 markers per cell type (*top_n2*) led to a) optimal results when used with DSA; b) similar results as using the *bottom_50p_AveExpr* or *bottom_50p_logFC* with ordinary linear regression strategies; c) worse results than random when used with penalized regression strategies (lasso, ridge, elastic net, DCQ) and CIBERSORT.

For *all* markers across each dataset, we took a closer look at the fold-change distribution for both the cell type where they were initially found as marker (highest fold change) and the fold-change differences among all other cell types. Using the threshold values used to select a gene as marker, we computed the percentage of those that could also be considered markers for a secondary cell type (values between parentheses in the boxplots below). For the five datasets included in the benchmark, 7–38% of the markers were not specific (exclusive) for only one cell type (see Supplementary Fig. 2).

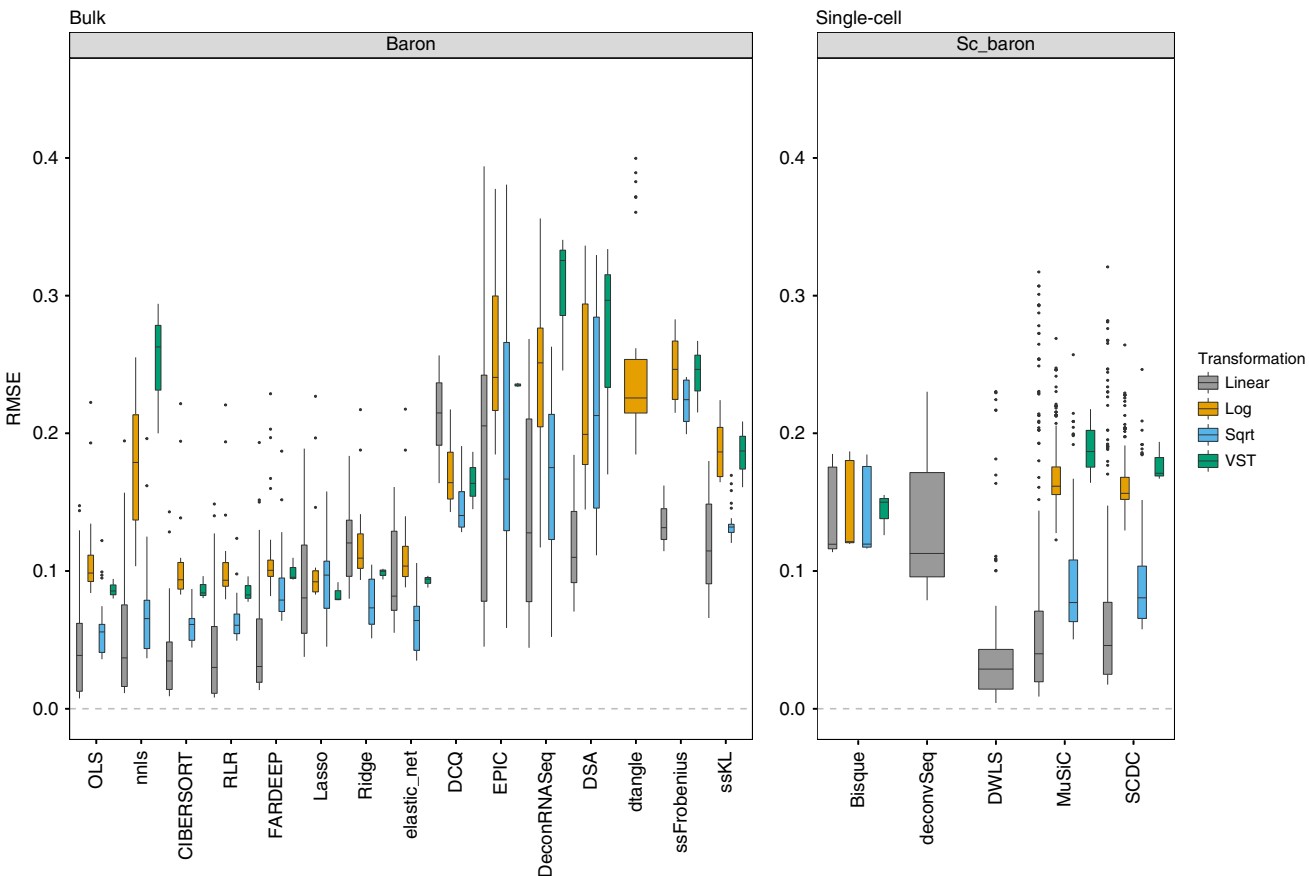

**Fig. 2 Impact of the data transformation on the deconvolution results.** RMSE values between the known proportions in 1000 pseudo-bulk tissue mixtures from the Baron dataset (pool size = 100 cells per mixture) and the predicted proportions from the different bulk deconvolution methods (left) and those using scRNA-seq data as reference (right). Each boxplot contains all normalization strategies that were tested in combination with a given method.

**Effect of removing cell types from the reference matrix.** Based on the results from all the analyses thus far, we decided to evaluate the impact of removing cell types with the data in linear scale and using all available markers (*all* marker selection strategy). Furthermore, we selected nnls and CIBERSORT as representative top-performing bulk deconvolution methods and DWLS and MuSiC as top-performing deconvolution methods that use scRNA-seq data as reference. To also be able to evaluate the impact of the normalization strategy, we included a representative sample of normalization strategies that result in small RMSE and high Pearson correlation values (see Fig. 3 and Supplementary Figs. 10–12): column, median ratios, none, TMM and TPM for nnls and CIBERSORT; column, scater, scran, none, TMM and TPM for DWLS and MuSiC.

We assessed the impact of removing a specific cell type by comparing the absolute RMSE values between the ideal scenario where the reference matrix contains all the cell types present in the pseudo-bulk mixtures (leftmost column in Figs. 5a, b and 6a, b (with gray label: none); Supplementary Figs. 16, 17) and the RMSE values obtained after removing one cell type at a time from the reference (all other gray labels).

We then focussed on those cases where the median absolute RMSE values between the results using the complete reference matrix (depicted as none in Figs. 5a, b and 6a, b) and all other scenarios where a cell type was removed, increased at least 2-fold. In the PBMC dataset (Fig. 5a, b), removing CD19+, CD34+, CD14+ or NK cells had an impact on the computed T-cell proportions (between a three and six-fold increase in the median absolute RMSE values, both in bulk deconvolution methods and those using scRNA-seq data as reference). The GSE81547 dataset

(Fig. 6a, b) shows that removing acinar cells has a dramatic impact in all other cell type proportions. Supplementary Figs. 14 and 15 showed the results for Baron and E-MTAB-5061 datasets, respectively. None of the method and normalization combinations was able to provide accurate cell type proportion estimates when the reference was missing a cell type.

To investigate whether the proportion of the omitted cell type was re-distributed equally among all remaining cell types or only among those that are transcriptionally most similar, we computed pairwise Pearson correlation values between the expression profiles of the different cell types (Figs. 5c, d and 6c, d). Figure 5c, d shows that CD14+ monocytes were mostly correlated with dendritic cells (Pearson = 0.85 when computing pairwise correlations on the reference matrix containing only marker genes and 0.94 when using the complete expression profiles from all cell types, respectively) and Fig. 5a, b shows that, when removing CD14+ monocytes, the highest RMSE value was found in dendritic cells. Figure 6c, d shows that acinar cells are not correlated with any other cell type (Pearson values close to zero with all other cell types) and Fig. 6a, b shows that, when removing acinar cells, all cell type proportions estimates have higher RMSE values compared to the case where no cell type is missing (none, leftmost panel).

For the Baron dataset (Supplementary Figs. 14 and 18): the removal of ductal cells (highest correlation with quiescent stellate and endothelial cells) led to highest RMSE values for both quiescent stellate and endothelial cells, while the removal of endothelial cells (mostly correlated with quiescent stellate, beta and ductal cells) led to the highest RMSE values for quiescent, ductal and beta cells. For the E-MTAB-5061 dataset

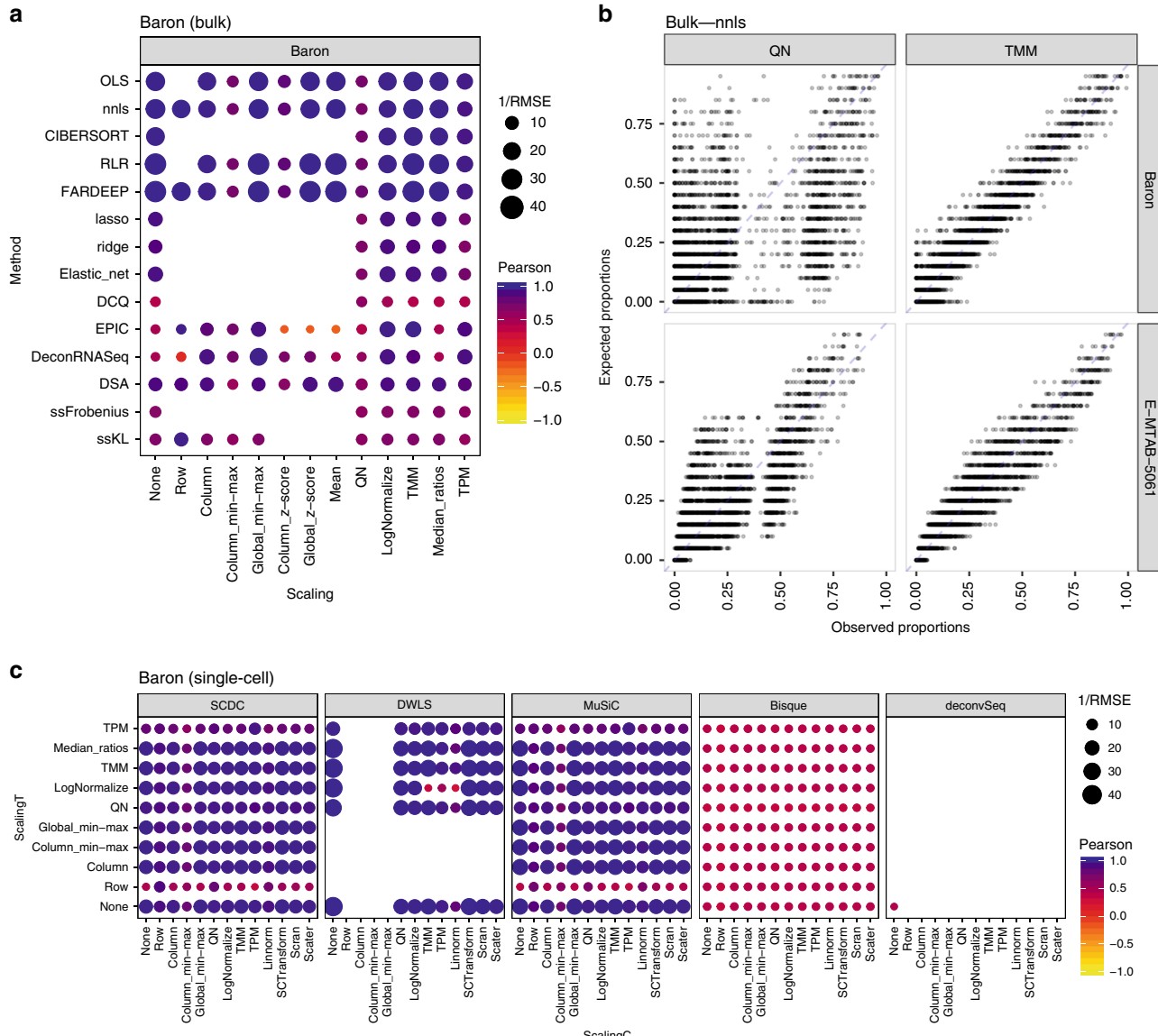

**Fig. 3 Combined impact of data normalization and methodology on the deconvolution results.** RMSE and Pearson correlation values between the expected (known) proportions in 1000 pseudo-bulk tissue mixtures in linear scale (pool size = 100 cells per mixture) and the output proportions from the different bulk deconvolution methods (**a**) and those using scRNA-seq data as reference (**c**). The darker the blue and the higher the area of the circle represents higher Pearson and lower RMSE values, respectively. **b** Scatter plot showing the impact of the normalization strategy (TMM versus quantile normalization (QN)) comparing the expected proportions (*y*-axis) and the results obtained through computational deconvolution using nnls (*x*-axis) for Baron and E-MTAB-5061 datasets. Empty locations represent combinations that were not feasible (see Supplementary Notes).

(Supplementary Figs. 15 and 19): no cell type is correlated to one another and removing any cell type from the reference matrix led to distorted proportions for all other cell types.

**Deconvolution of real bulk heterogeneous samples**. In contrast to the thousands of artificial pseudo-bulk mixtures across five datasets used in the previous sections, we used nine human bulk PBMCs samples from Finotello et al.[27] for which cell type proportions were measured by flow cytometry. We considered these proportions as the gold standard against which both bulk deconvolution methods and those able to use scRNA-seq data as reference could be evaluated. To note, it was not possible to evaluate MuSiC and SCDC because the 10x scRNA-seq data used as reference came from only one individual. Hence, only DWLS, deconvSeq, and BisqueRNA were tested. See "Computational

framework for the evaluation of deconvolution pipelines with real RNA-seq data" ("Methods") and Table 1 for more details.

Regarding bulk deconvolution methods: robust regression methods (RLR, FARDEEP) and support vector regression (CIBERSORT) consistently showed the smallest RMSE and highest Pearson correlation values (Fig. 7a). Similarly, DWLS performed best among the deconvolution methods that use scRNA-seq data as input (Fig. 7b).

**Discussion**

Using both Pearson correlation and RMSE values as measures of the deconvolution performance, we comprehensively evaluated the combined impact of four data transformations, sixteen scaling/normalization strategies, eight marker selection approaches and twenty different deconvolution methodologies on five different scRNA-seq datasets. These datasets encompass three

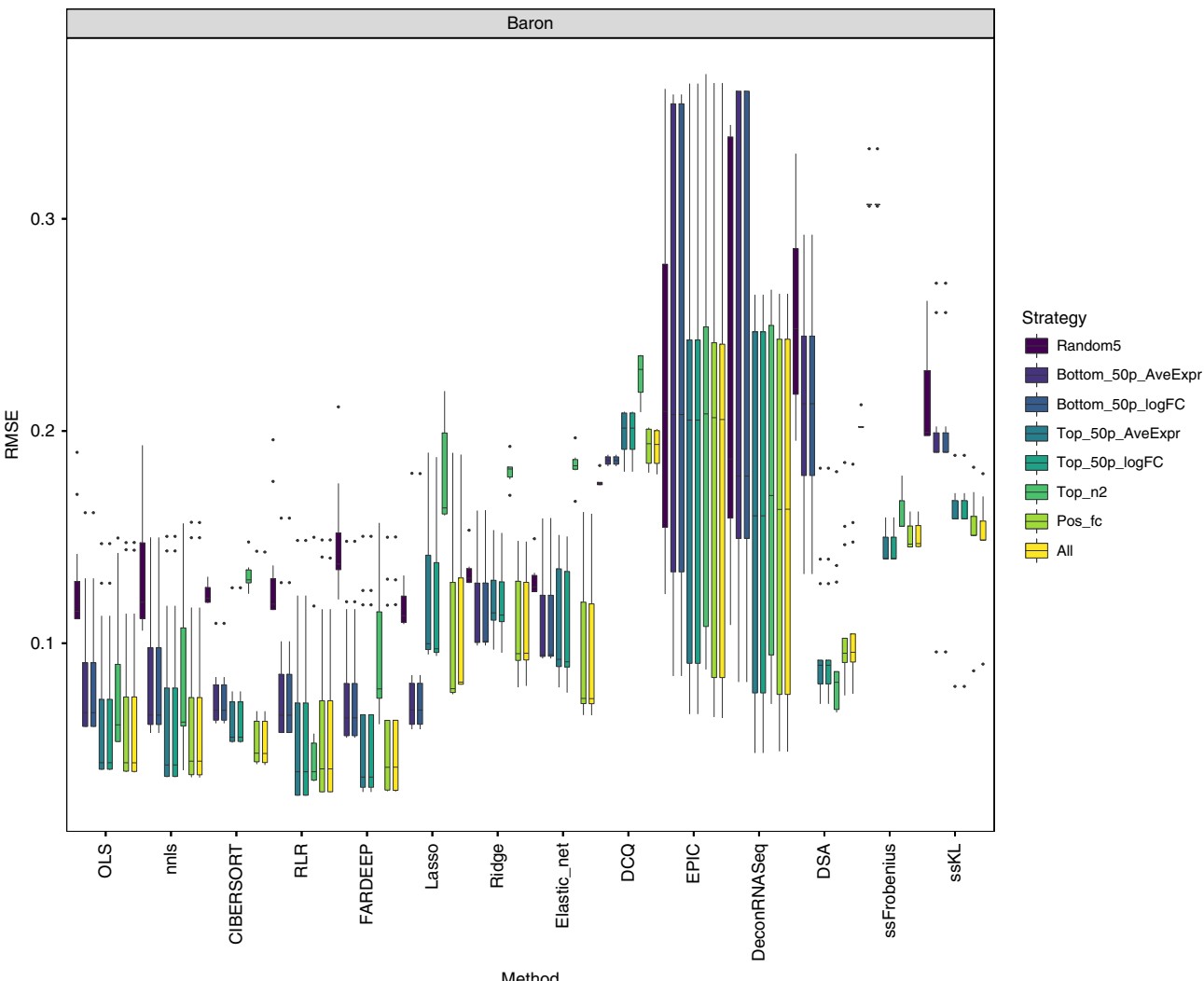

**Fig. 4 Impact of the marker selection on the deconvolution results.** RMSE values between the expected (known) proportions in 1000 pseudo-bulk tissue mixtures (linear scale; pool size = 100 cells per mixture) and the output proportions from the Baron dataset, using eight different marker selection strategies. Each boxplot contains all normalization strategies that were tested in combination with a given marker strategy across the different bulk deconvolution methods.

different biological sample types (human pancreas, kidney, and peripheral blood mononuclear cells) and four different sequencing protocols (CEL-Seq, Smart-Seq 2, Microwell-Seq, and GemCode Single-Cell 3′). Additionally, we assessed the impact of using different number of cells when making the pseudo-bulk mixtures and the impact of removing cell types from the reference matrix that were actually present in the mixtures.

Even though the five scRNA-seq datasets used throughout this manuscript encompass different sequencing protocols that led to hundred-fold differences in the number of reads sequenced per cell (Table 1), our findings were consistent regardless of the dataset being evaluated or the number of cells used to make the pseudo-bulk mixtures (Supplementary Figs. 6–12). Given the limited number of cells available per dataset and the scarcity of publicly available datasets with similar health status, sequencing platform, and library preparation protocol to validate our results, some cells were used in more than one mixture and each dataset was split into training and testing (50%:50%), meaning that cells from one individual were present both in training and test sets but a given cell was only present in one split. Nevertheless, while the different datasets (except PBMCs) contain cells from more than one individual (=inherent inter-sample variability), we

observed meaningful differences between cell types rather than by individual (Supplementary Fig. 24). Additionally, we generated scenarios where cells from a given individual were used only in one split (training or test) by assigning half of the samples to each split prior to selecting the cells based on the cell type. These led to slightly higher RMSE and lower Pearson correlation values compared to those where cells from one individual were present in both splits, but the same conclusions hold true in both analyses (Supplementary Figs. 25–26).

Both cell type proportions on their own (e.g., at baseline level, before any treatment has started) and changes in cell type composition upon drug treatment or a viral infection are relevant and can be assessed through computational deconvolution. For instance, patients with high levels of tumor-infiltrating lymphocytes have been found to respond better to immune checkpoint inhibitors (immunotherapy)[28] and changes in diverse immune cell types were found in mice lungs during the course of influenza infection[29]. In principle, the performance of a computational deconvolution framework should be independent of the experimental set up where it is applied. However, we acknowledge that the data included in our benchmark did not directly evaluate the latter scenario.

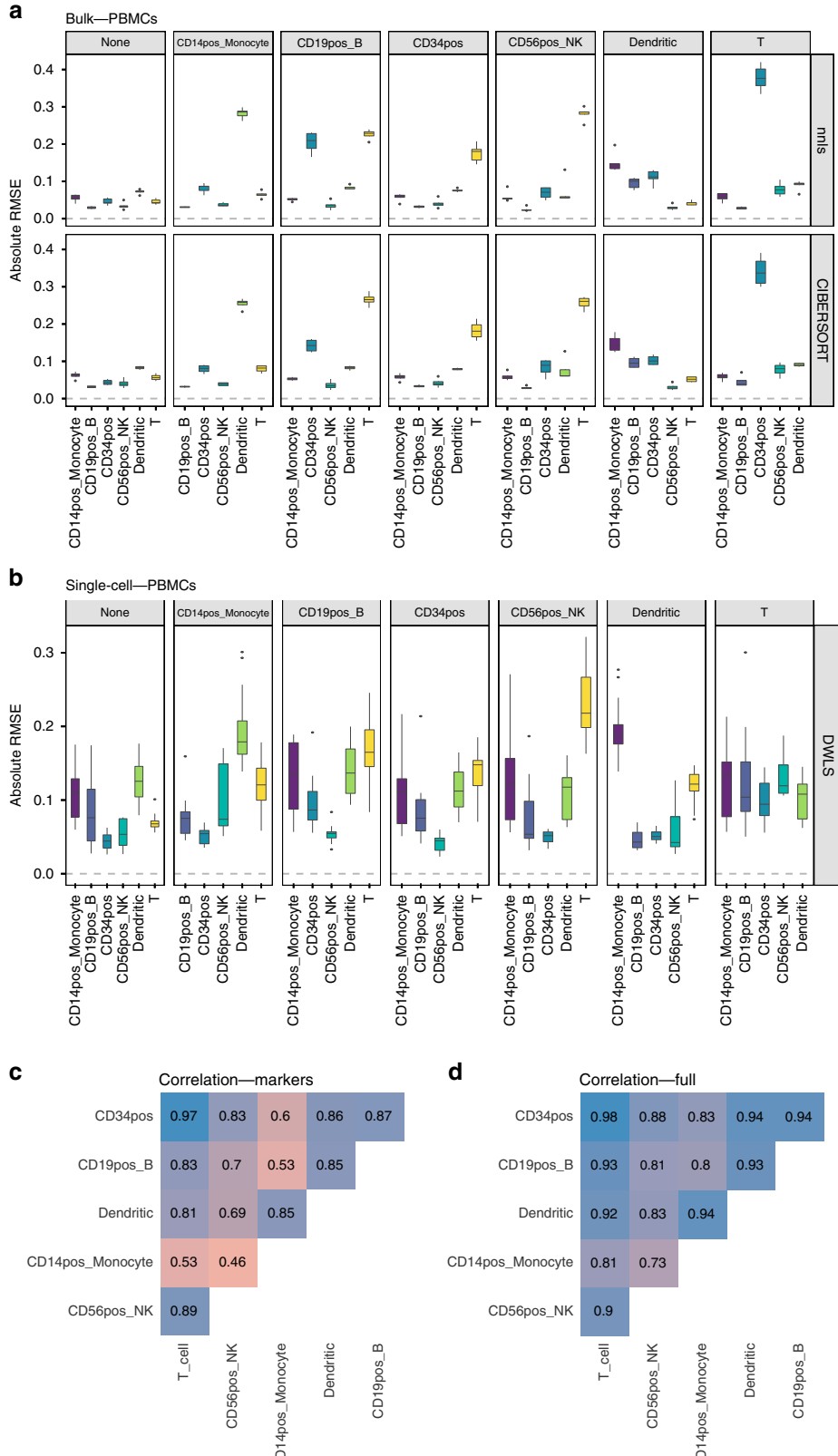

**Fig. 5 Effect of cell type removal on the deconvolution results for the PBMCs dataset (100-cell mixtures; linear scale). a** Results using bulk deconvolution methods (nnls and CIBERSORT); **b** results with deconvolution methods using scRNA-seq data as reference (only DWLS because the data comes from only one individual). **c** Pairwise Pearson correlation values between expression profiles for the different cell types, using a subset of the reference matrix containing only the markers used in the bulk deconvolution; **d** pairwise Pearson correlation values between complete expression profiles for the different cell types. In **a**, **b**, each gray column represents a specific cell type removed. Each data point conforming a boxplot represents a different scaling/normalization strategy used.

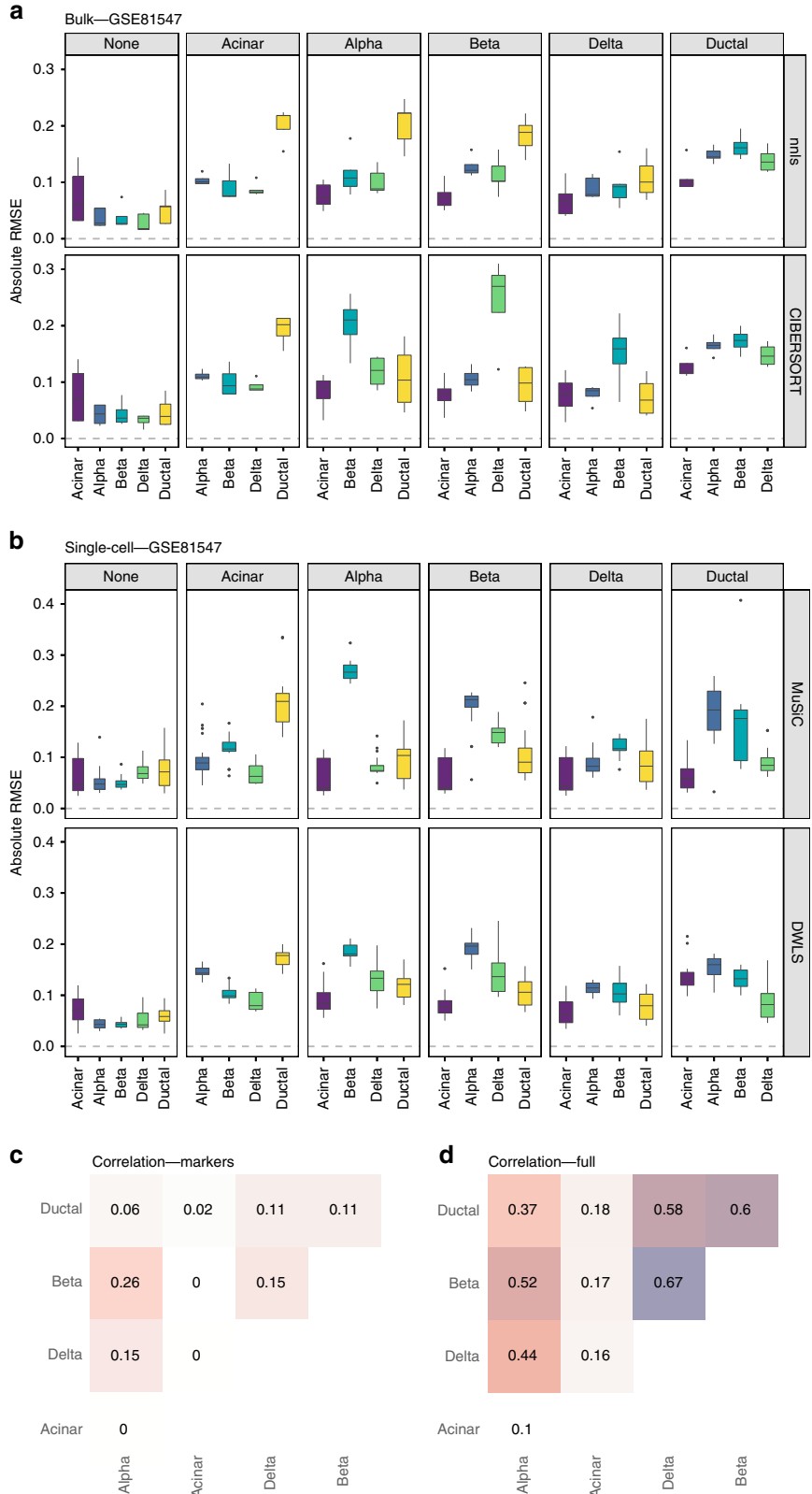

**Fig. 6 Effect of cell type removal on the deconvolution results for the GSE81547 dataset (100-cell mixtures; linear scale). a** Results using bulk deconvolution methods (nnls and CIBERSORT); **b** results with deconvolution methods using scRNA-seq data as reference (MuSiC and DWLS). **c** Pairwise Pearson correlation values between expression profiles for the different cell types, using a subset of the reference matrix containing only the markers used in the bulk deconvolution; **d** pairwise Pearson correlation values between complete expression profiles for the different cell types. In **a**, **b**, each gray column represents a specific cell type removed. Each data point conforming a boxplot represents a different scaling/normalization strategy used.

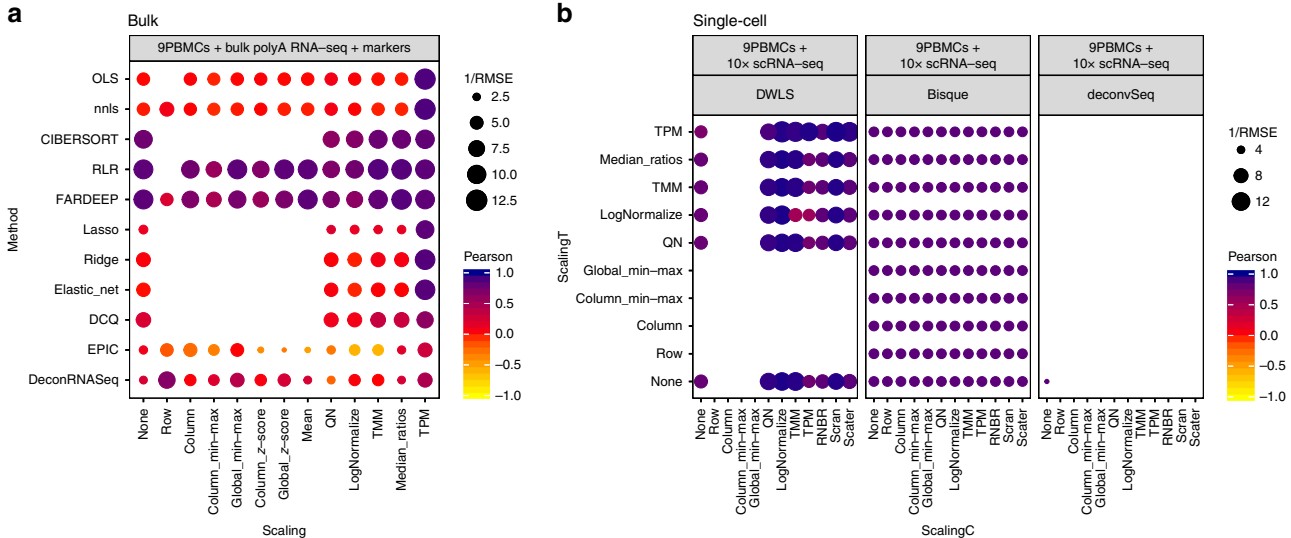

**Fig. 7 Deconvolution performance on nine human PBMC bulk samples.** With **a** bulk deconvolution methods; **b** deconvolution methods using scRNA-seq as reference.

The logarithmic transformation is routinely included as a part of the pre-processing of omics data in the context of differential gene expression analysis[30,31], but Zhong and Liu[6] showed that it led to worse results than performing computational deconvolution in the linear (un-transformed) scale. The use of the expression data in its linear form is an important difference with respect to classical differential gene expression analyses, where statistical tests assume underlying normal distributions, typically achieved by the logarithmic transformation[32]. Silverman et al.[33] showed that using log counts per million with sparse data strongly distorts the difference between zero and non-zero values and Townes et al.[34] showed the same when log-normalizing UMIs. Tsoucas et al.[23] showed that when the data was kept in the linear scale, all combinations of three deconvolution methods (DWLS, QP, or SVR) and three normalization approaches (LogNormalize from Seurat, Scran or SCnorm) led to a good performance, which was not the case when the data was log-transformed. Here, we assessed the impact of the log transformation on both full-length and tag-based scRNA-seq quantification methods and confirmed that the computational deconvolution should be performed on linear scale to achieve the best performance.

Data scaling or normalization is a key pre-processing step when analysing gene expression data. Data scaling approaches transform the data into bounded intervals such as [0, 1] or [−1, +1]. While being relatively easy and fast to compute, scaling is sensitive to extreme values. Therefore, other strategies that aim to change the observations so that they follow a normal distribution (= normalization) may be preferred. Importantly, these normalizations typically do not result in bounded intervals. In the context of transcriptomics, normalization is needed to only keep true differences in expression. Normalizations, such as TPM aim at removing differences in sequencing depth among the samples. An in-depth overview of normalization methods and their underlying assumptions is presented in Evans et al.[35]. Vallania et al.[11] assessed the impact of standardizing both the bulk and reference expression profiles into z-scores prior to deconvolution, which is performed by CIBERSORT but not in other methods. They observed high pairwise correlations between the estimated cell type proportions with and without standardizing the data, suggesting a neglectable effect. However, a high Pearson correlation value is not always synonym of a good performance. As already pointed out by Hao et al.[36], high Pearson correlation values can arise when the proportion estimations are accurate

(low RMSE values) but also when the proportions differ substantially (high RMSE values), making the correlation metric alone not sufficient to assess the deconvolution performance. Both for bulk deconvolution methods and those that use scRNA-seq data as reference, our analyses show that the normalization strategy had little impact (except for EPIC, DeconRNASeq, and DSA bulk methods). Of note, quantile normalization (QN), an approach used by default in several deconvolution methods (e.g., FARDEEP, CIBERSORT), consistently showed sub-optimal performance regardless of the chosen method.

In general, the use of all data at hand (i.e., in supervised strategies) leads to better results than unsupervised or semi-supervised approaches. However, in other contexts different from computational deconvolution (e.g., automatic cell identification[37]), it has been shown that incorporating prior knowledge into the models does not improve the performance. Furthermore, there are situations where cell-type specific expression profiles are not readily available and supervised methodologies cannot be used. For these reasons, we included ssFrobenius and ssKL in our benchmarking, two semi-supervised non-negative matrix factorization methods to perform bulk gene expression deconvolution. They led to higher RMSE and lower Pearson correlation values than most supervised methodologies (except DCQ and dtangle; Fig. 2 and Supplementary Fig. 6), highlighting the positive impact of incorporating prior knowledge (in the form of cell-type specific expression profiles) in the field of computational deconvolution. In any case, results from supervised and semi-supervised methodologies should be interpreted separately.

Schelker et al.[38] and Racle et al.[26] showed that the origin of the expression profiles had also a dramatic impact on the results, revealing the need of using appropriate cell types coming from niches similar to the bulk being investigated. Hunt et al.[39] showed that a good deconvolution performance was achieved if the markers being used were predominantly expressed in only one cell type and with the expression in other cell types being in the bottom 25%. Monaco et al.[40] showed similar conclusions when the reference matrix was pre-filtered by removing markers with small log fold change between the first and second cell types with highest expression. In our analyses, markers were selected based on the fold change with respect to the cell type with the second-highest expression. Therefore, the pre-filtering proposed by Hunt et al. and Monaco et al. was already implicitly done. Furthermore, when subsetting the markers based on their average gene expression or fold

changes, those in the top fifty percent led to smaller RMSEs compared to those in the bottom fifty percent (Fig. 4).

Wang et al.[24] explored the effect of removing one immune cell type at a time from the reference matrix on the estimation accuracy using artificial bulk expression of six pancreatic cell types (alpha, beta, delta, gamma, acinar, and ductal) and removing one cell type from the single-cell expression dataset. They observed that, when a cell type was missing in the reference matrix, MuSiC, NNLS, and CIBERSORT did not produce accurate proportions for the remaining cell types. Gong and Szustakowski[20] also investigated this issue by performing a first deconvolution using DeconRNASeq, then removing the least abundant cell population from the reference/basis matrix, and finally repeating the deconvolution with the new matrix. They observed an uneven redistribution of the signal and observed that some initial proportions became smaller. Moreover, Schelker et al.[38] investigated this phenomenon by looking at the correlation coefficient between the results obtained with the complete reference matrix and the results removing one cell type at a time.

We performed similar analyses for four deconvolution methods (two bulk and two using scRNA-seq data as reference) and eleven normalization strategies (five for bulk, six for single-cell) on three single-cell human pancreas and one PBMC dataset, keeping the data in linear scale. We observed both cases where the choice of normalization strategy had no impact and other cases where it did. Interestingly, the removal of specific cell types did not affect all other cell types equally. Both bulk deconvolution methods and those using scRNA-seq data as reference showed similar trends when removing specific cell types. However, there were some discrepancies in the RMSE values (e.g., removal of beta cells had a substantial impact on the proportions of delta cells but CIBERSORT showed three times higher RMSE values compared to either nnls, MuSiC or DWLS (Fig. 6a, b and Supplementary Fig. 17)). This may be explained by the fact that for bulk deconvolution methods, we removed both the cell type expression profile and its marker genes from the reference matrix whereas for those where scRNA-seq data was used as reference, only the cells from the specific cell type were excluded, without applying extra filtering on the genes (MuSiC, SCDC) or because a different signature was internally built (DWLS).

Schelker et al. found that B cell and dendritic cell proportions were affected by removing macrophages or monocytes whereas NK cell proportions were affected by removing T cells. Sturm et al., also reported the impact of removing CD8+ T cells on NK cell proportions. Our results on the PBMC dataset agree with those from Schelker et al. and Sturm et al. but also include novel insights: removing CD19+ B-cells, CD34+, CD14+ monocytes or NK cells had an impact on the computed T-cell proportions and removing CD19+ B-cells, CD56+ NK or T cells had an impact on CD34+ cell proportions.

Furthermore, we found a direct association between the correlation values among the cell types present in the mixtures and the effect of removing a cell type from the reference matrices. Specifically, we hypothesize that: (a) removing a cell type that is barely or completely uncorrelated (Pearson < 0.2) to all other cell types remaining in the reference matrix has a dramatic impact in the cell type proportions of all other cell types; (b) removing a cell type that was strongly positively correlated (Pearson > 0.6) with one or more cell types still present in the reference matrix leads to distorted estimates for the most correlated cell type(s). The correlation between different cell types is a direct manifestation of their relatedness in a cell-type ontology/hierarchy: the closer the cell types in the hierarchy, the higher the correlation between their expression profiles. The cell-type relationship based on the hierarchy is a good qualitative predictor of the population, which will be most affected when removing a cell type from the reference matrix.

EPIC[26] shows a first attempt in alleviating this problem by considering an unknown cell type present in the mixture. Nevertheless, this is currently restricted to cancer, using markers of non-malignant cells that are not expressed in cancer cells.

In conclusion, when performing a deconvolution task, we advise users to: (a) keep their input data in linear scale; (b) select any of the scaling/normalization approaches described here with exception of row scaling, column min-max, column $z$-score or quantile normalization; (c) choose a regression-based bulk deconvolution method (e.g., RLR, CIBERSORT or FARDEEP) and also perform the same task in parallel with DWLS, MuSiC or SCDC if scRNA-seq data is available; (d) use a stringent marker selection strategy that focuses on differences between the first and second cell types with highest expression values; (e) use a comprehensive reference matrix that include all relevant cell types present in the mixtures.

Finally, as more scRNA-seq datasets become available in the near future, its aggregation (while carefully removing batch effects) will increase the robustness of the reference matrices being used in the deconvolution and will fuel the development of methodologies similar to SCDC, which allows direct usage of more than one scRNA-seq dataset at a time.

## Methods

**Dataset selection and quality control**. Five different datasets coming from different single-cell isolation techniques (FACS and droplet-based microfluidics) and encompassing both full-length (Smart-Seq2) and tag-based library preparation protocols (3′-end with UMIs) were used throughout this article (see Table 1). After removing all genes (rows) full of zeroes or with zero variance, those cells (columns) with library size, mitochondrial content or ribosomal content further than three median absolute deviations (MADs) away were discarded. Next, only genes with at least 5% of all cells (regardless of the cell type) with a UMI or read count greater than 1 were kept.

Finally, we retained cell types with at least 50 cells passing the quality control step and, by setting a fixed seed and taking into account the number of cells across the different cell types (pooling different individuals when possible; thereby including inherent inter-sample variability), each dataset was further split into balanced training and testing datasets (50%:50% split) with a similar distribution of cells per cell type.

Regarding E-MTAB-5061: cells with *not_applicable*, *unclassified* and *co-expression_cell* labels were excluded and only cells coming from six healthy patients (non-diabetic) were kept.

After quality control, we made two-dimensional t-SNE plots for each dataset. When adding colored labels both by cell type and donor (Supplementary Fig. 24), the plots showed consistent clustering by cell type rather than by donor, indicating an absence of batch effects.

**Generation of reference matrices for the deconvolution**. Using the training splits from the previous section, the mean count across all individual cells from each cell type was computed for each gene, constituting the original (un-transformed and un-normalized) reference matrix ($C$ in equation (1) from section "Computational deconvolution: formulation and methodologies") and were used as input for the bulk deconvolution methods described in that section.

For the deconvolution methods that use scRNA-seq data as reference and for the marker selection step, the training subsets were used in their original single-cell format, whereas a mean gene expression collapsing step (= mean expression value across all cells of the same cell type) was required to generate the reference matrices used in the bulk deconvolution methods.

**Cell-type specific marker selection**. TMM normalization (edgeR package[41]) was applied to the original (linear) scRNA-seq expression datasets and limma-voom[42] was used to find out marker genes. Only genes with positive count values in at least 30% of the cells of at least one group were retained. Among the retained ones, those with absolute fold changes greater or equal to 2 with respect to the second cell type with highest expression and BH adj $p$-value < 0.05 were kept as markers in all three pancreatic datasets. Since the kidney and PBMC datasets contained more closely related cell types, the fold-change threshold was lowered to 1.8 and 1.5, respectively.

Once the set of markers was retrieved, the following approaches were evaluated: (i) *all*: use of all markers found following the procedure described in the previous paragraph; (ii) *pos_fc*: using only markers with positive fold-change (= over-expressed in cell type of interest; negative fold-change markers are those with small expression values in the cell type of interest and high values in all the others); (iii) *top_n2*: using the top 2 genes per cell type with the highest log fold-change; (iv)

**Table 2 Evaluation of deconvolution pipelines with real mixtures.**

|  | Heterogeneous input | Reference matrix | Marker information | Expected proportions |
|---|---|---|---|---|
| Bulk deconvolution methodologies | 9 human bulk PBMCs | Bulk RNA-seq of monocytes, B cells, natural killer, dendritic cells and T cells (SRA accession from Supplementary Table 1) | Markers for monocytes, B cells, natural killer, dendritic cells and T cells obtained from 10x scRNA-seq data | Flow cytometry measurements |
| Deconvolution methods using scRNA-seq as reference | 9 human bulk PBMCs | 10× scRNA-seq data excluding CD34 + cells | No prior information | Flow cytometry measurements |

Scenarios used to assess the performance of the different deconvolution methods using bulk RNA-seq from nine PBMCs samples.

**Table 3 Detailed description of different scaling/normalization approaches used in the benchmarking.**

| Scaling/normalization method | Single-cell specific | Output containing negative values | Output bounded in [0,1] interval | Reference |
|---|---|---|---|---|
| Column-wise (= Total count or library size normalization) | No | No | Yes | 59 |
| Column min-max | No | No | Yes | 60 |
| Column $z$-score | No | Yes | No | 61 |
| Row-wise | No | No | Yes | 62 |
| Global min-max | No | No | Yes | 60 |
| Global $z$-score | No | Yes | No | 61 |
| Quantile normalization (QN) | No | No | No | 63 |
| Upper quartile (UQ) | No | No | No | 64 |
| Transcripts per million (TPM) | No | No | No | 65 |
| Trimmed mean of M-values (TMM) | No | No | No | 66 |
| LogNormalize | No | No | No | 67 |
| Median of ratios | No | No | No | 13 |
| Scran | Yes | No | No | 16 |
| Scater | Yes | No | No | 68 |
| Linnorm | Yes | No | No | 69 |
| SCTransform (RNBR) | Yes | No | No | 15 |

*top_50p_logFC*: top 50% of markers (per CT) based on log fold-change; (v) *bottom_50p_logFC*: bottom 50% of markers based on log fold-change; (vi) *top_50p_AveExpr*: top 50% of markers based on average gene expression (baseline expression); vii) *bottom_50p_AveExpr*: low 50% based on average gene expression; (viii) *random5*: for each cell type present in the reference, five genes that passed quality control and filtering were randomly selected as markers.

**Generation of thousands of artificial pseudo-bulk mixtures.** Using the testing datasets from the quality control step, we generated matrices containing 1000 pseudo-bulk mixtures (matrix $T$ in equation (1) from "Computational deconvolution: formulation and methodologies") by adding up count values from the randomly selected individual cells. The minimum number of cells used to create the pseudo-bulk mixtures (pool size) for each of the five datasets was 100 and the maximum possible number was determined by the second most abundant cell type (rounded down to the closest hundred, to avoid non-integer numbers of cells), resulting in $n = 100$, 700, and 1200 for Baron; $n = 100$, 300, and 400 for PBMCs; $n = 100$ and 200 for GSE81547; $n = 100$ and 200 for the kidney dataset and $n = 100$ for E-MTAB-5061.

For the human pancreas and PBMC datasets, each (feasible) pseudo-bulk mixture was created by randomly (uniformly) selecting the number of cell types to be present (between 2 and 5) and their identities, followed by choosing the cell type proportion assigned to each cell type (enforcing a sum-to-one constraint) among all possible proportions between 0.05 and 1, in increasing intervals of 0.05. For the kidney data, the number of cell types present was randomly (uniformly) selected between 2 and 8 (eight being the maximum number of cell types possible), followed by selecting the cell type proportion assigned to each cell type (enforcing a sum-to-one constraint) among all possible proportions between 0.01 and 0.99. Finally, once the number of cells to be picked up from specific cell types was determined, the cells were randomly selected without replacement (= a given cell can only be present once in a mixture).

**Evaluation of deconvolution pipelines with real RNA-seq data.** We downloaded and processed raw poly(A) RNA-seq (single-end) data of nine human (bulk) PBMCs samples from Finotello et al.[27] for which cell type proportions were measured by flow cytometry (assumed gold standard; see Supplementary Table 2

and "Processing poly(A) RNA-seq of nine human bulk PBMCs samples" in Supplementary Notes). Furthermore, we used scRNA-seq data from PBMCs (10× Genomics; see Table 1) and bulk RNA-seq for B cells, monocytes, myeloid dendritic cells, natural killer and T cells (see Supplementary Table 1).

Importantly, we acknowledge two limitations in this set-up: (i) the nine PBMCs include a measurement for neutrophils (2.45%–5.05%) while such cell type was not present in the 10x scRNA-seq data; (ii) The 10× scRNA-seq data contained CD34+ cells whereas the nine PBMCs did not include information for such cell type.

Therefore, to establish an unbiased assessment for both bulk deconvolution methods and those that use scRNA-seq data as reference, we excluded CD34+ cells from the 10× scRNA-seq data and did not use the bulk RNA-seq data for neutrophils in the reference matrix (see Table 2). Of note, flow cytometry proportions for T cells were computed as the sum of proportions of three different sub-populations ($T_{regs}$, CD8+ and CD4+).

**Data transformation and normalization.** The next step is applying four different data transformations to: (i) the un-transformed and un-normalized reference matrix C; (ii) the un-transformed and un-normalized single-cell training splits and (iii) the un-transformed and un-normalized matrix T containing the 1000 pseudo-bulk mixtures.

Since count data from both bulk and scRNA-seq show the phenomenon of over-dispersion[41,43], the following data transformations were chosen: (a) leave the data in the original (linear) scale; (b) use the natural logarithmic transformation (with the log1p function in R[44]); (c) use the square-root transformation; (d) variance-stabilizing transformation (VST). The second and third are simple and commonly used transformations aiming at reducing the skewness in the data due to the presence of extreme values[31] and stabilizing the variance of Poisson-distributed counts[45], respectively. VST (using the varianceStabilizingTransformation from DESeq2) removes the dependence of the variance on the mean, especially important for low count values, while simultaneously normalizing with respect to library size[13].

Each transformed output file was further scaled/normalized with the approaches listed on Table 3. The mathematical implementation can be found at the original publications (*Ref* column) and in our GitHub repository. Due to the sparsity of the scRNA-seq matrices (most genes with zero counts), the UQ

normalization failed (all normalization factors were infinite or NA values) and thus was eventually not included in downstream analyses. TMM includes an additional step that uses the normalization factors to obtain normalized counts per million. LogNormalize and Linnorm include an additional exponentiation scale after normalization in order to transform the output data back into linear scale. Median of ratios can only be applied to integer counts in linear scale.

**Computational deconvolution: formulation and methodologies.** The deconvolution problem can be formulated as:

$$T = C \cdot P \tag{1}$$

(see Avila Cobos et al. [5] together with "Approximation of bulk transcriptomes as linear mixtures" and "Small impact of cell cycle in the deconvolution results" in Supplementary Notes), where $T$ = measured expression values from bulk heterogeneous samples; $C$ = cell type-specific expression values and $P$ = cell-type proportions. Specifically, $T$ represents the 1000 pseudo-bulk mixtures from "Generation of thousands of artificial pseudo-bulk mixtures" and C is the reference matrix from "Cell-type specific marker selection and generation of reference matrices for the deconvolution". In the context of this article, the goal is to obtain $P$ using $T$ and $C$ as input.

Fifteen bulk deconvolution methods a have been evaluated, including two traditional (ordinary least squares (OLS[21]) and non-negative least squares (NNLS[22])) and one weighted least squares method (EPIC[26]); two robust regression (FARDEEP[46], RLR[47]), one support-vector regression (CIBERSORT[9]) and four penalized regression (ridge, lasso, elastic net[48] and Digital Cell Quantifier (DCQ[29])) approaches; one quadratic programming (DeconRNASeq[20]), one method that models the problem in logarithmic scale (dtangle[39]) and three methods included in the CellMix R package:[19] Digital Sorting Algorithm (DSA[17]) and two semi-supervised non-negative matrix factorization methods (ssKL and ssFrobenius[18]). Furthermore, five deconvolution methods that use scRNA-seq as reference have been evaluated: deconvSeq[49], MuSiC[24], DWLS[23], Bisque[50] and SCDC[25]. We refer the reader the original publications and our Github repository (github.com/favilaco/deconv_benchmark) for details about their implementation.

**Measures of deconvolution performance.** Changes in memory were assessed with the mem_change function from the pryr package[51] and the elapsed time was measured with the proc.time function (both functions executed in R v.3.6.0).

We computed both the Pearson correlation values and the root-mean-square error (RMSE) between cell type proportions from thousands of pseudo-bulk mixtures with known composition and the output from different deconvolution methods for each combination of data transformation, scaling/normalization choice, and deconvolution method. Higher Pearson correlation and low RMSE values correspond to a better deconvolution performance.

**Evaluation of missing cell types in the reference matrix C.** For every cell type removed, the deconvolution was applied only to mixtures where the missing cell type was originally present. For bulk deconvolution methods, the marker genes of the cell type that was removed from the reference were also excluded (methods using scRNA-seq data as reference did not require a priori marker information).

**Reporting summary.** Further information on research design is available in the Nature Research Reporting Summary linked to this article.

## Data availability

The five publicly available datasets used in this article can be found at their respective sources: Baron: https://www.ncbi.nlm.nih.gov/geo/query/acc.cgi?acc=GSE84133 (Specifically: GSM2230757, GSM2230758, GSM2230759 and GSM2230760 for human pancreatic islands). GSE81547, E-MTAB-5061, PBMCs: https://support.10xgenomics.com/single-cell-gene-expression/datasets/1.1.0/fresh_68k_pbmc_donor_a, kidney.HCL: https://figshare.com/articles/HCL_DGE_Data/7235471, see Table 1 for more details.

## Code availability

Source code can be found at https://github.com/favilaco/deconv_benchmark.

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

## Acknowledgements

We would like to acknowledge Evan Benn, Derrick Lin, Vikkitharan Gnanasambandapillai, and Manuel Sopena Ballesteros (Garvan-Weizmann Centre for Cellular Genomics) for their IT support; Lucía Lorenzi (Ghent University) for testing the public Github repository and Annelien Morlion (Ghent University) for her help processing publicly available poly-A RNA-sequencing data. This work was supported by the European Union's Horizon 2020 research and innovation programme under grant agreement 668858, a Special Research Fund scholarship from Ghent University (BOF. DOC.2017.0026.01) and a scholarship for a long stay abroad (V440318N) from the Fund for Scientific Research Flanders (FWO).

## Author contributions

F.A.C. conducted all the analyses and wrote the paper. J.A.H. contributed to the quality control assessment of the scRNA-seq data. K.D.P., P.M., and J.E.P. provided guidance on the analyses and interpretation of results. All authors reviewed, edited, and approved the final paper.

## Competing interests

The authors declare no competing interests.
