## [Peer Review File · Nature Communications]

Reviewers' comments:

Reviewer #1 (Remarks to the Author):

Cobol et al. perform a comparison of different methods that can be applied for computational deconvolution of transcriptome data.

The manuscript repeats many analyses that have been performed before for evaluation of deconvolution methods. This limits the novelty and perhaps makes it more suitable as a review or benchmarking manuscript in a specialist bioinformatics journal.

The choices of how to present the results obtained is questionable. For example, a substantial section of the Results is dedicated to demonstrating that normalization results in different results. Although the authors acknowledge in Discussion that this has already been addressed (in 2012), they should acknowledge that this is not new when the results are presented (for proper contextualization of the result) and should probably consider dedicating less space to results, such as this, that are not novel.

It does not really make sense to compare performance of the semi-supervised and supervised methods – the scenarios in which they would be used and the input data are different. The semi-supervised methods can be used to make inferences about sample composition when only the identity and not the expression levels in the reference cell types of the marker genes are known. It would not be reasonable to use the semi-supervised methods if gene expression data in the reference cell types is available.

The datasets used for evaluation are critical for a manuscript such as this. It was not clear to me that the training and testing data was split in such a way that cells from the same individual could not be in both the testing and training data (given that one dataset was derived from just a single individual, I assume that this was not the case). In a real setting, there are sources of variability between the reference profiles and the target bulk expression data that are not accounted for in this strategy (e.g. inter-individual variation and experimental/batch variation between the reference and target data). This could cause the authors to over-estimate the performance of the deconvolution methods. Also the number of mixtures created appears to have been far greater than the number of experiments from which the single-cell expression data was derived, again potentially resulting in under-estimation of the variability with which the deconvolution methods would have to contend in a real setting.

Specific comments

The authors open the results with some benchmarking of computational requirements. They argue that computational constraints imposed by the methods may be important for non-specialist users, but it was not clear that the tasks they performed had computational requirements within an order of magnitude of what would be available on a personal computer, at least in terms of memory (the maximum value on the y-axis of Fig. 1, left panels, was only around 100 MB). Therefore it was not clear that evaluation of memory use, as represented in this figure, was particularly relevant.

The authors state that "Remarkably, no method and normalization combination was able to provide accurate cell type proportion estimates when the reference missed a cell type." It was not clear to me why this finding was remarkable. Would the authors not expect that missing cell types to distort the proportion of all of the remaining cell types, especially since the proportions of all cell types must sum to one? Fig. 6 displays absolute RMSE on the y-axis. Would it be better to present the differences in relative proportions of the remaining cell types after one cell type has been removed?

"taking into account the number of cells across the different cell types, each dataset was further split

into "training" and "testing" datasets with a similar distribution of cells per cell type" Was this a 50:50 split or split in some other proportion?

Reviewer #2 (Remarks to the Author):

In the manuscript the authors present a comprehensive benchmarking study of several computational methods for quantifying cell type composition from cell mixture measurements (the computational deconvolution task). The benchmark is based on generating synthetic datasets, based on single cell RNASeq profiles derived from publicly available datasets, where cells of different types have been characterized. The main novelty of the study with respect to existing benchmarks is comprehensive testing of various data normalization and transformation strategies together with deconvolution methods. The number of tested approaches is also larger than it was done before.

I found the study design convincing, and providing such a benchmark is a timely contribution. The manuscript is well-written and all messages are clearly explained.

I have few remarks which I believe could further clarify certain aspects of the study and may be make the message stronger.

The study represents comprehensive benchmarking of the methods which are supervised (use reference profiles at the deconvolution step) by their nature and deals only with transcriptomic data. I think it should be highlighted it in the title. Also, 'benchmarking of deconvolution' sounds strange, I suggest to add 'methods'.

The naming 'single-cell deconvolution methods' is misleading because what is deconvoluted is not a single-cell dataset. A better designation for this method should be provided.

Splitting a single cell dataset into training and test part is a straightforward but more realistic validation should be taking reference profiles from one dataset and generate the pseudo-bulk profiles from an independent dataset. Is such a validation envisageable?

The experiments with removing a cell type which exists in the mixture but not in the reference matrix are insightfull but remain too empirical. Can it be analyzed analytically for a general case, with a simple statistical mixture modeling? I think this will improve the universality of the conclusions.

The figures for RMSE and correlations have been computed for a pseudo-bulk collection of profiles, where the known proportions are generated using a grid of values (in other words, all possible combinations). Therefore, one difficulty connected to the analysis of real-life data (colinearity problem) has not been taken into account. How the results of the benchmarking change if there exists a clear correlation structure between different cell type abundancies aka real-life data? For example, what if the known proportions of two cell types are always similar in a sample set? If some methods work not good in such situation (e.g., this can be anticipated for the simplest OLS) then the median RMSE can change dramatically.

The strategy of generating synthetic data was linear mixing of few reference profiles. This approach is

straightforward

but it seems to have strong limitations. My suggestion to the authors is to make an effort to demonstrate that real bulk transcriptomes can be approximated as such linear mixtures. For example, if the reference profiles are chosen as explanatory factors, how much variance they explain in real bulk transcriptomes?

The benchmark does not investigate the effect of cell type-independent factors which can affect transcriptome (e.g. proportions of proliferating cells). Can it be considered?

Using only 5 cell types in the benchmark seems to be a strong limitation of the study. Can the authors clarify why more cell types can not be used in principle? Also, three datasets in the benchmark are from the human pancreas, why it does not create a bias?

I understand the difficulty of using real-life data in such a benchmarking, however, there are datasets where certain cell types have some expected ranges (e.g., some type of immune cells should be absent in certain tissues). Can this be used to make a stronger point for the comparison of deconvolution methods in real datasets?

The meaning and scale of RMSE as a measure of deconvolution quality is unclear. Can the authors provide some arguments why, for example, RMSE lower than 0.05 is a good deconvolution result as they claim.

The Figure 2 does not seem informative or necessary to me. The timing of most of the methods and memory use seem to be reasonable and not limiting for practical studies. My suggestion is to move it to Supplementary. Also, I think it is better to report the computation time per sample.

For the "all markers" strategy, it would be interesting to know if there were any overlap in genes as it was stated before that previous analyses speculated that the lack of specificity in markers explained some bad results for proportion estimations.

In the abstract, the authors claim that 'the choice of the reference marker signatures is far more important than the method itself'. I think this can not be stated as a well-established fact. How many 'attempts' confirm this? The problem is that in many situations it is not easy to test all combinations of markers and methods, so I do not think this statement was 100% demonstrated.

Of note, I did not want to access the private GitHub provided in the manuscript for anonymity reasons, therefore I made my evaluations based on the public repository https://github.com/favilaco/deconv_benchmark. Currently, the code accompanying the study is organized in one single R script. Since the code generating the synthetic data can be reused in other studies with different methods or strategies to generate the synthetic data, I suggest to reorganize the code into 1) synthetic data generator part, 2) various method caller part following unified interface, 3) proper benchmarking part.

Test run for the Single cell example gave me RMSE=0.0821538, Pearson=0.934689 versus documented RMSE=0.06977075, Pearson=0.9527703.

It is relatively minor but still 17% RMSE difference.

The code was run on a Mac with all dependency packages installed and updated, using R version 3.6.1.

Where might these differences come from? Can the OS/platform have an impact on the benchmarking result in general and

can it alter the quality of some analyses? Has this code been tested on various platforms to check for reproducibility?

Minor remarks:

Supplementary Figure 10 C,D are not readable. They can be zoomed in a PDF of course but I would suggest to make them printable.

In Figure 2 the panels are not described in their order in the figure legend. This makes it difficult to digest.

In Line 178, I am confused by the terminology used. I discover that supervised methods assumes expression values

for markers (weights) while the methods using sets of marker genes are called semi-supervised.

So, MCPCounter would be a semi-supervised method (I do not agree with this).

Note also than any unsupervised method also uses sets of marker genes for interpreting the meaning of factors/clusters.

Figure 6c is difficult to read in print. Can it be improved?

Lines 395-396 the statement about 'mean collapsing' is not clear to me.

Lines 422-424 : considerations about possible pool sizes are not clear to me.

Reviewer #3 (Remarks to the Author):

In this manuscript, Cobos and colleagues evaluate deconvolution methods (and associated preprocessing strategies) using simulations involving pseudo-bulk single-cell RNA-seq data. They generate some measures of performance for each method from datasets and provide some recommendations for obtaining best results. The manuscript is clear and addresses an interesting problem; however, there are some deficiencies with the evaluation framework that make it challenging to extrapolate the results to real-life application of deconvolution methods. I have listed my concerns in more detail below.

1) I understand that the authors' evaluation is based purely on pseudo-bulk single-cell RNA-seq profiles. However, the methods under discussion are intended to operate on actual bulk RNA-seq profiles. There is a subtle but important distinction here, as bulk RNA-seq remains quite different from the library preparation technology used in single-cell experiments. I list a few such differences below:

- Many bulk protocols use random primers compared to the poly-A approach that dominates single-cell protocols, driving differences in capture efficiency across genes.
- Bulk protocols tend to have much fewer rounds of PCR, reducing amplification biases.
- Bulk protocols can also afford to undergo RNA purification and other clean-up steps (e.g., ribosomal depletion).
- Bulk protocols preserve differences in RNA content between multiple cell types in a mixed

population, whereas such differences are difficult to recover in single-cell data without spike-ins.
- Bulk data is typically full-length without UMIs, so using pseudo-bulk UMI counts is especially unrealistic.

All of these factors can skew the observed expression profile such that the bulk counts are not simply the sum of single-cell counts.

Why do I mention this? Well, testing methods on pseudo-bulk profiles gives an overly optimistic impression of the performance of each method. The simulated mixtures have exactly the same properties as the reference profiles, which seems divorced from real applications of deconvolution methods. The current evaluation does not capture the robustness of each method to technical differences between protocols such as, e.g., changes in capture efficiency for key marker genes.

It would seem more meaningful to construct the mixtures using bulk RNA-seq from pure populations (e.g., after FACS sorting). There are a number of datasets where such data are available; ImmGen or the Database of Immune Cell Expression come to mind. The evaluation would then focus on the accuracy of using single-cell reference profiles to deconvolve those bulk-derived mixtures in an accurate yet robust manner.

2) The authors focus on the RMSE between the predicted and true cell type proportions to determine method performance. This is hard to interpret; I would instead like to know the error relative to the magnitude of the proportion. For example, a method with an RMSE of 0.05 for a cell type might not be so bad if that type's true proportion is 0.5, but it would be much less appealing if the true proportion is 0.01.

On a related note, I would say that the absolute proportion holds little biological meaning by itself. Much more interesting is the difference in proportions between conditions, e.g., does a particular cell type increase in abundance before and after drug treatment? To this end, a useful evaluation would look at whether relative differences in the true cell type proportions are preserved in the deconvolution output.

3) It was not clear to me what proportions were used to construct the mixtures. I would hope that the authors created mixtures with increasingly rare populations to examine the performance of each method for recovering correct values when the true proportions of each cell type span a wide range of values.

4) The authors mention that the methods operate best on non-transformed data. This is nice to know but is this not obvious? The bulk mixture is constructed (in silico or in the lab) by pooling expression profiles on the original scale, so it seems clear that the best approach would be to perform deconvolution on the same scale. It would be helpful to have some elaboration on why this is not a trivial result.

5) It would be necessary to show that the differences in the methods have some kind of impact on real data analyses. I do not expect the authors to evaluate the methods on real data or to generate a gold-standard mixture experiment; I just want to see that any differences in behavior observed in the simulations are reflected in real data. For example, if one method struggles to detect rare populations in simulations, we should also see a dearth of small proportions when applied on real data.

Point-by-point response to the referees' comments

We would like to thank the reviewers for thoroughly reading our manuscript and their constructive suggestions, as we believe they have improved our manuscript. We have adjusted our manuscript according to the reviewers' comments and the present document provides detailed answers (blue) to each comment (in black).

Reviewer #1 (Remarks to the Author):

Cobos *et al.* perform a comparison of different methods that can be applied for computational deconvolution of transcriptome data.

1) The manuscript repeats many analyses that have been performed before for evaluation of deconvolution methods. This limits the novelty and perhaps makes it more suitable as a review or benchmarking manuscript in a specialist bioinformatics journal. The choices of how to present the results obtained is questionable. For example, a substantial section of the Results is dedicated to demonstrating that normalization results in different results. Although the authors acknowledge in Discussion that this has already been addressed (in 2012), they should acknowledge that this is not new when the results are presented (for proper contextualization of the result) and should probably consider dedicating less space to results, such as this, that are not novel.

We respectfully disagree with this comment. The main novelty of our study with respect to other benchmarks (e.g. Sturm *et al.*, 2019¹) is the comprehensive evaluation of many commonly used data scaling/normalization and transformation strategies together with (semi-)supervised deconvolution methods.

In the introduction we describe several studies that have addressed different factors affecting the deconvolution results, but each of them only focused on one or two individual aspects at a time. The reference from 2012 (Zhong, Y. & Liu, Z. *Gene expression deconvolution in linear space. Nat. Methods* 9, 8–9 (2012)²) only showed the importance of data transformation on the deconvolution results, but did not consider normalization. The choice of normalization method has been shown to dramatically impact the results of differential gene expression pipelines^{1,3,4} and has also been assessed in the context of cell clustering⁵. Nevertheless, it has not been comprehensively evaluated yet in the context of computational deconvolution, becoming one of the main goals of our manuscript.

2) It does not really make sense to compare performance of the semi-supervised and supervised methods – the scenarios in which they would be used and the input data are different. The semi-supervised methods can be used to make inferences about sample composition when only the identity and not the expression levels in the reference cell types of the marker genes are known. It would not be reasonable to use the semi-supervised methods if gene expression data in the reference cell types is available.

We agree with the reviewer. In most cases, the use of all data at hand (i.e. supervised strategies) leads to better results than unsupervised or semi-supervised counterparts. However, in other contexts different from computational deconvolution (e.g. automatic cell identification⁶), it has been shown that incorporating prior knowledge into the models does not improve the performance. Therefore, we found worthwhile to include the assessment of ssFrobenius and ssKL, two non-negative matrix factorization variants that use prior knowledge (sets of marker genes) to perform gene expression deconvolution. These methods perform worse (higher median RMSE values) than many (but not all) alternative methods (Figure 2, comparison with DCQ or dtangle).

3) The datasets used for evaluation are critical for a manuscript such as this. It was not clear to me that the training and testing data was split in such a way that cells from the same individual could not be in both the testing and training data (given that one dataset was derived from just a single individual, I assume that this was not the case). In a real setting, there are sources of variability between the reference profiles and the target bulk expression data that are not accounted for in this strategy (e.g. inter-individual variation and experimental/batch variation between the reference and target data). This could cause the authors to over-estimate the performance of the deconvolution methods. Also the number of mixtures created appears to have been far greater than the number of experiments from which the single-cell expression data was derived, again potentially resulting in under-estimation of the variability with which the deconvolution methods would have to contend in a real setting.

We thank the reviewer for raising this important point (which was also raised as comment #3 from Reviewer 2). Given the limited number of cells available per dataset, we acknowledge that some cells were used in more than one mixture. Furthermore, we also acknowledge that the ideal situation would be a second (independent) dataset for testing. Unfortunately, scRNA-seq datasets with similar health status, sequencing platform and library preparation protocol are currently scarce. Therefore, we split each dataset into both training and testing (50%:50%) as described below.

Nevertheless, we observed no differences by individual (“batch”) but by cell type (see Supplementary Figure 24).

The following changes have been made (in red):

Section “Dataset selection and quality control” (Methods):

[...] Finally, we retained cell types with at least 50 cells passing the quality control step and, by setting a fixed seed and taking into account the number of cells across the different cell types (pooling different individuals when possible; thereby including inherent inter-sample variability), each dataset was further split into balanced “training” and “testing” datasets (50%:50% split) with a similar distribution of cells per cell type.

Discussion section:

[...] Given the limited number of cells available per dataset and the scarcity of publicly available datasets with similar health status, sequencing platform and library preparation protocol to validate our results, some cells were used in more than one mixture and each dataset was split into training and testing (50%:50%). Nevertheless, while the different datasets (except PBMCs) contain cells from more than one individual (= inherent inter-sample variability), we observed meaningful differences between cell types rather than by individual (Supplementary Figure 24).

Specific comments

4) The authors open the results with some benchmarking of computational requirements. They argue that computational constraints imposed by the methods may be important for non-specialist users, but it was not clear that the tasks they performed had computational requirements within an order of magnitude of what would be available on a personal computer, at least in terms of memory (the maximum value on the y-axis of Fig. 1, left panels, was only around 100 MB). Therefore it was not clear that evaluation of memory use, as represented in this figure, was particularly relevant.

We agree with the reviewer. We have now moved this figure from the main manuscript to Supplemental material (Supplementary Figure 5), have removed the first paragraph from the section “Different normalization and methodology combinations have different memory requirements and time consumption” (Results) and have added this sentence:

Running time and memory use of the different deconvolution methods is summarized in Supplementary Figure 5.

5) The authors state that “Remarkably, no method and normalization combination was able to provide accurate cell type proportion estimates when the reference missed a cell type.” It was not clear to me why this finding was remarkable. Would the authors not expect that missing cell types to distort the proportion of all of the remaining cell types, especially since the proportions of all cell types must sum to one?

Fig. 6 displays absolute RMSE on the y-axis. Would it be better to present the differences in relative proportions of the remaining cell types after one cell type has been removed?

We agree with the reviewer that this choice of words was not appropriate, since it is indeed an expected result. The deconvolution methods have either implicit or explicit sum-to-one constraints when computing the cell type proportions (Supplementary Table 3), meaning that these will sum up to one regardless of the amount of cell types in the reference matrix. Therefore, we have removed “Remarkably” from the paragraph.

Following the proposal for Figure 6 (re-labelled as Figure 5 in the updated version), we have generated an additional plot for each dataset by removing the leftmost square (ideal situation where all cell types were present in the reference matrix) and use a new OY axis: RMSE fold change with respect to the ideal situation. The horizontal dashed line at FC = 1 represents no changes observed, and values greater than one represent a detriment:

Supplementary Figure 16 – PBMCs dataset (linear scale): RMSE fold change with respect to the ideal situation (=where all cell types in the mixtures were also present in the reference matrix) for both bulk deconvolution methods (top panel) and those that use scRNA-seq data as reference (bottom panel). The horizontal dashed line at FC = 1 represents no changes observed, and values greater than one represent a detriment. Each grey column represents a specific cell type removed. Each data point conforming a boxplot represents a different scaling/normalization strategy used.

All of these figures have been included as Supplementary Material (Supplementary Figures 16-19) and have been referred to in the main manuscript as follows (red):

We assessed the impact of removing a specific cell type by comparing the absolute RMSE values between the ideal scenario where the reference matrix contains all the cell types present in the pseudo-bulk mixtures (leftmost column in Figures 5a-b and 6a-b (with grey label “none”); **Supplementary Figures 16-17**) [...] For the baron dataset (Supplementary Figure 14 **and 18**): [...] For the E-MTAB-5061 dataset (Supplementary Figure 15 **and 19**)

6) “taking into account the number of cells across the different cell types, each dataset was further split into “training” and “testing” datasets with a similar distribution of cells per cell type” Was this a 50:50 split or split in some other proportion?

We apologize for the confusion here, it was indeed a 50:50 split. For details please see the answer to comment #3 above.

Reviewer #2 (Remarks to the Author):

In the manuscript the authors present a comprehensive benchmarking study of several computational methods for quantifying cell type composition from cell mixture measurements (the computational deconvolution task). The benchmark is based on generating synthetic datasets, based on single cell RNASeq profiles derived from publicly available datasets, where cells of different types have been characterized. The main novelty of the study with respect to existing benchmarks is comprehensive testing of various data normalization and transformation strategies together with deconvolution methods. The number of tested approaches is also larger than it was done before.

I found the study design convincing, and providing such a benchmark is a timely contribution. The manuscript is well-written and all messages are clearly explained.

I have few remarks which I believe could further clarify certain aspects of the study and may make the message stronger.

1) The study represents comprehensive benchmarking of the methods which are supervised (use reference profiles at the deconvolution step) by their nature and deals only with transcriptomic data. I think it should be highlighted in the title. Also, 'benchmarking of deconvolution' sounds strange, I suggest to add 'methods'.

Including the word “methods” will only represent a portion of the factors being evaluated (transformation, normalization, method, marker selection and cell type composition of the reference matrix). Therefore, we have opted for the word “pipelines” and the new title reads as:

“Benchmarking of cell type deconvolution pipelines for transcriptomics data”

2) The naming 'single-cell deconvolution methods' is misleading because what is deconvoluted is not a single-cell dataset. A better designation for this method should be provided.

“single-cell deconvolution methods” has been substituted by “deconvolution methods using single-cell RNA-sequencing (scRNA-seq) data as reference” throughout the manuscript.

3) Splitting a single cell dataset into training and test part is a straightforward but more realistic validation should be taking reference profiles from one dataset and generate the pseudo-bulk profiles from an independent dataset. Is such a validation envisageable?

We thank the reviewer for raising this fair point (which was also raised as comment #3 from Reviewer 1). Given the limited number of cells available per dataset, we acknowledge that some cells were used in more than one mixture. Furthermore, we also acknowledge that the ideal situation would be a second (independent) dataset for testing. Unfortunately, scRNA-seq datasets with similar health status, sequencing platform and library preparation protocol are currently scarce. Therefore, we split each dataset into both training and testing (50%:50%) as described below.

Nevertheless, we observed no differences by individual (“batch”) but by cell type (see Supplementary Figure 24).

The following changes have been made (in red):

Section “Dataset selection and quality control” (Methods):

[...] Finally, we retained cell types with at least 50 cells passing the quality control step and, by setting a fixed seed and taking into account the number of cells across the different cell types (pooling different individuals when possible; thereby including inherent inter-sample variability), each dataset was further split into balanced “training” and “testing” datasets (50%:50% split) with a similar distribution of cells per cell type.

Discussion section:

[...] Given the limited number of cells available per dataset and the scarcity of publicly available datasets with similar health status, sequencing platform and library preparation protocol to validate our results, some cells were used in more than one mixture and each dataset was split into training and testing (50%:50%). Nevertheless, while the different datasets (except PBMCs) contain cells from more than one individual (= inherent inter-sample variability), we observed meaningful differences between cell types rather than by individual (Supplementary Figure 24).

4) The experiments with removing a cell type which exists in the mixture but not in the reference matrix are insightful but remain too empirical. Can it be analyzed analytically for a general case, with a simple statistical mixture modeling? I think this will improve the universality of the conclusions.

The use of statistical mixture modelling (e.g. Gaussian Mixture Models (GMMs)) on heterogeneous samples (e.g. the pseudo-bulk mixtures used in our manuscript) would “guess” the number of cell types intrinsically hidden. Assuming that the guess is correct (which will not always be the case), we could check whether the reference matrix to be used in the deconvolution is complete or is missing one or more cell types. Nevertheless, even in the scenarios where the predicted number of cell types coincide with the number of cell types in the reference matrix, an extra step would be needed to determine whether the cell types included in the reference are appropriate or not (for instance, by computing the proportion of the variance in the dependent variable that is predictable from the independent variable(s); see answer to comment 6 below). Therefore, we feel that such analytical analysis for a general case is extremely hard for such benchmark.

Nevertheless, the correlation between different cell types is a direct manifestation of their relatedness in a cell-type ontology/hierarchy: the closer the cell types in the hierarchy, the higher the correlation between their expression profiles. Hypothetically, when removing a given cell type from the reference, we would expect that the absolute root-mean squared error (RMSE) increased for the most similar cell type(s) based on their cell-hierarchy organization (see hematopoietic lineage scheme below).

The following table shows the relationship between the empirical observations and the previous hypothesis:

Removed cell type	Empirical: (Observed) most affected cell type (highest RMSE, Figure 5 from the main manuscript)	Hypothetical: (Expected) most affected cell type (closest in hierarchy)
CD14+ Monocytes	Dendritic cells	Dendritic cells
CD19+ B cells	T cells	T cells and NK
Dendritic cells	CD14+ Monocytes	CD14+ Monocytes
T cells	CD56+ NK	CD56+ NK
CD56+ NK	T cells	T cells

As seen on the table, the cell-type relationship based on the hierarchy is a good qualitative predictor of the population which will be most affected when removing a cell type from the reference matrix (highest RMSE). The CD34+ population is expected to be widely heterogeneous because the FACS enrichment will capture both lymphoid and myeloid progenitor cells in a trajectory (rather than discrete/differentiated populations). Therefore, it is expected to be correlated to all remaining cells at some degree depending on the lymphoid/myeloid variance of progenitor cells affecting the expression values of the signature genes.

We have added the following to the Discussion (in red):

Furthermore, we found a direct association between the correlation values among the cell types present in the mixtures and the effect of removing a cell type from the reference matrices. Specifically, we hypothesize that: a) removing a cell type that is barely or completely uncorrelated (Pearson < 0.2) to all other cell types remaining in the reference matrix has a dramatic impact in the cell type proportions of all other cell types; b) removing a cell type that was strongly positively correlated (Pearson > 0.6) with one or more cell types still present in the reference matrix leads to distorted estimates for the most correlated cell type(s). The correlation between different cell types is a direct manifestation of their relatedness in a cell-type

ontology/hierarchy: the closer the cell types in the hierarchy, the higher the correlation between their expression profiles. The cell-type relationship based on the hierarchy is a good qualitative predictor of the population which will be most affected when removing a cell type from the reference matrix.

5) The figures for RMSE and correlations have been computed for a pseudo-bulk collection of profiles, where the known proportions are generated using a grid of values (in other words, all possible combinations). Therefore, one difficulty connected to the analysis of real-life data (colinearity problem) has not been taken into account. **How the results of the benchmarking change if there exists a clear correlation structure between different cell type abundancies aka real-life data? For example, what if the known proportions of two cell types are always similar in a sample set?** If some methods work not good in such situation (e.g., this can be anticipated for the simplest OLS) then the median RMSE can change dramatically.

As the reviewer has pointed out, the proportions were generated using all possible combinations without artificially introducing an intrinsic correlation structure. We only assessed the impact of correlated cell types in our initial version of the manuscript (collinearity problem in the reference matrix).

We feel that the generation of all possible correlations between each pair of cell types across all datasets and cell pool sizes is unfeasible, but we have investigated such question using the Baron dataset and generated 100 pseudo-bulk mixtures for each of the 10 fixed mixes depicted in the heatmap below (left hand side; each column represents one mix).

That way we artificially imposed the proportion of alpha cells to always be the same as for delta cells or twice as big. We perform the deconvolution using nnls and TPM on the linear scale (one of the top combinations across all datasets) and the results are depicted on the figure below (right hand side; delta cells in red, alpha cells in blue; the other three cell types in black), where we did not notice an improvement/worsening of the computed proportions for those two cell types compared to the other three also present in the mixtures.

Nevertheless, this is obviously a limited analysis that would require a full assessment for all datasets and deconvolution frameworks (transformation + normalization + method) tested throughout the manuscript. As we feel this huge effort will not add significant insights into the problem, we opted for not including this analysis in the manuscript.

6) The strategy of generating synthetic data was linear mixing of few reference profiles. This approach is straightforward but it seems to have strong limitations. My suggestion to the authors is to make an effort to demonstrate that real bulk transcriptomes can be approximated as such linear mixtures. For example, if the reference profiles are chosen as explanatory factors, how much variance they explain in real bulk transcriptomes?

We downloaded publicly available raw poly(A) RNA-sequencing data (Illumina HiSeq 2000; paired-end) from the Sequence Read Archive (SRA; NCBI) of two bulk PBMCs from healthy donors and constituent (bulk) B cells, Monocytes, Natural Killer and T cells together with the corresponding cell type proportions measured by flow cytometry (Supplementary File 3C from <https://elifesciences.org/articles/26476/figures#supp3>):

SRA accession	Cell type	Donor
SRR1740034	B cells	Donor: HD30
SRR1740038	myeloid DC	Donor: HD30
SRR1740042	Monocytes	Donor: HD30
SRR1740046	Neutrophils	Donor: HD30
SRR1740050	NK cells	Donor: HD30
SRR1740054	PBMC	Donor: HD30
SRR1740058	T cells	Donor: HD30
SRR1740062	B cells	Donor: HD31
SRR1740066	myeloid DC	Donor: HD31
SRR1740070	Monocytes	Donor: HD31
SRR1740074	Neutrophils	Donor: HD31
SRR1740078	NK cells	Donor: HD31
SRR1740082	PBMC	Donor: HD31
SRR1740086	T cells	Donor: HD31

We aligned the RNA-seq reads against the human genome (Homo sapiens; Ensembl v91; GRCh38) with STAR v2.6.0c and the output .bam files from the previous step were used as input for HTSeq v0.11.0, resulting in a final matrix with gene counts.

Since we have the three components for the problem $T = C \cdot P$ (see section “Computational deconvolution: formulation and methodologies”; T made of “PBMCs”; C composed of “B cells, Monocytes, Myeloid DCs, Neutrophils, NK and T cells”; P from flow cytometry), we used R^2 (proportion of the variance in the dependent variable that is predictable from the independent variable(s) and also indicates the goodness of fit of the model) as proxy to demonstrate that real bulk transcriptomes can be approximated as linear mixtures of the constituent cell types (=explanatory factors).

The output was $R^2_{\text{HD30}} = 0.962$ and $R^2_{\text{HD31}} = 0.942$, meaning that >94% of the variance in real bulk PBMC transcriptomes is explained by choosing a linear mixing of its constituent CTs and therefore, we conclude that bulk transcriptomes can therefore be approximated as such linear mixtures.

We have included this answer as a new section entitled "Approximation of bulk transcriptomes as linear mixtures" in the Supplemental material and added the following in the main manuscript (red):

The deconvolution problem can be formulated as $T = C \cdot P(I)^7$ (see "approximation of bulk transcriptomes as linear mixtures" [...] in Supplementary Methods

7) The benchmark does not investigate the effect of cell type-independent factors which can affect transcriptome (e.g. proportions of proliferating cells). Can it be considered?

The impact of cell cycle phases has been largely overlooked in the deconvolution field in general and current deconvolution frameworks assume cell-type specific markers to be insensitive or invariant to this factor.

To investigate the impact of the cell cycle on deconvolution, we used the Baron et al. and PBMC single-cell RNA-seq datasets as input for the "cyclone" function developed by Scialdone *et al.*⁸ as part of the "scrn" package (R statistical programming language), which contains a pre-trained set of human marker gene pairs that allows the classification of cells into different cell cycle phases.

After cells were initially classified into G1, S and G2M, we binarized the cells into "S" (proliferating) and "non-S" (= composed of G1 and G2M; non-proliferating) and, for each of the ratios 0:100 / 25:75 / 50:50 / 75:25 / 100:0 (% of S: % non-S), we artificially generated 100 pseudo-bulk mixtures of 100 cells each to evaluate the deconvolution framework "linear scale + LogNormalize + nnls". The scatter plots below revealed, for both datasets, very small differences in RMSE and Pearson correlation values across the different ratios (Baron: $\Delta_{\text{RMSE}} = 0.06 - 0.04 = 0.02$; $\Delta_{\text{Pearson}} = 0.96 - 0.93 = 0.03$; PBMCs: $\Delta_{\text{RMSE}} = 0.05 - 0.03 = 0.02$; $\Delta_{\text{Pearson}} = 1 - 0.96 = 0.04$).

Of note, for the PBMC data, most of the cells used in the pseudo-bulk mixtures with 100:0 ratio are T-cells because other cell types were found to have fewer cells in cell cycle stage "S".

a

Baron – 100 mixtures.
Linear + LogNormalize + nnls

**b**

PBMCs – 100 mixtures.
Linear + LogNormalize + nnls

Supplementary Figure 1 – Small impact of cell cycle stage in the deconvolution results on a) Baron; b) PBMCs datasets, respectively. The number in each gray rectangle depicts the percentage of cells in S phase and, for each percentage, 100 pseudo-bulk mixtures were assessed with nnls (data in linear scale followed by LogNormalize).

We have included this answer as a new section entitled "small impact of cell cycle in the deconvolution results" in the Supplemental material and added the following in the main manuscript (red):

The deconvolution problem can be formulated as $T = C \cdot P(l)^T$ (see "approximation of bulk transcriptomes as linear mixtures" and "small impact of cell cycle in the deconvolution results" in Supplementary Methods), where T = measured expression values from bulk heterogeneous samples;

8) Using only 5 cell types in the benchmark seems to be a strong limitation of the study. Can the authors clarify why more cell types can not be used in principle?

We agree with the reviewer, and the only reason for choosing five was that it was the smallest number we could assess for each of the four datasets initially included in the benchmark (see Table 1 in the main manuscript). We have not repeated the pseudo-bulk mixtures we used for the initial four datasets included in the benchmark, but we have removed such constraint and have randomly (and uniformly) generated mixtures up to 8 cell types (maximum possible; with more than 100 cases per mix: mix of 2, 3, ... , 8 cell types) for the new kidney dataset included in the manuscript (see answers to your remarks 9 and 16 regarding the new results included).

We have also added the following (in red) to the manuscript (section "Generation of thousands of artificial pseudo-bulk mixtures):

For the kidney dataset, the number of cell types present was randomly (uniformly) selected between 2 and 8 (=maximum number of cell types possible) [...]

9) Also, three datasets in the benchmark are from the human pancreas, why it does not create a bias?

This is a fair point. With the initial version of the manuscript we showed that, even though the four single-cell RNA-seq datasets (3 pancreatic, 1 PBMCs) encompassed different sequencing protocols that led to hundred-fold differences in the number of reads sequenced per cell (Table 1), our findings were consistent regardless of the dataset being evaluated or the number of cells used to make the pseudo-bulk mixtures.

Nevertheless, we downloaded additional single-cell RNA-seq data for human kidney from Han et al., 2020⁹ (https://figshare.com/articles/HCL_DGE_Data/7235471). There were three adult samples available with five cell types in common, but following the previous remark (show results with more than 5 cell types), we decided to choose two that maximize the number of common cell types: Adult 2 and Adult 4, with eight cell types in common. We generated 1000 pseudo-bulk mixtures (100 and 200 cells per mixture; 2 to 8 cell types; proportions from 0.01 to 0.99 (see "Generation of thousands of artificial pseudo-bulk mixtures" for more details) and computed RMSE and Pearson correlation values in the same manner as we did for the pancreatic and PBMC datasets.

A) and C) represent the results for the Baron dataset (pancreas) for bulk deconvolution methods and methods using single-cell RNA-seq as reference, respectively. B) and D) are the results for the kidney dataset.

Supplementary Figure 12 – Similar Pearson correlation and RMSE values between pancreas (baron dataset, a) and c) and kidney (b) and d)) for both bulk deconvolution methods (a-b) and those that use scRNA-seq as reference (c-d).

Based on this analysis, we conclude that the results are similar across human pancreas, PBMCs and kidney tissue-types.

We have added the figure to Supplementary material (Supplementary Figure 12) and have adapted the main manuscript as follows (in red):

Introduction:

*The performance is assessed by means of Pearson correlation and root-mean-square error (RMSE) values between the cell type proportions computed by the different deconvolution methods (P_C ; computed proportions; Figure 1) and known compositions (P_E ; expected proportions) of a thousand pseudo-bulk mixtures from each of **five** different single cell RNA-sequencing datasets (three from human pancreas; **one from human kidney** and one from **human** peripheral blood mononuclear cells (PBMCs)).*

Methods:

Five different datasets coming from different single-cell isolation techniques [...]

Table 1 – Details of the five datasets used. [...]

Results (“Different combinations of normalization and deconvolution methodologies reveal important differences in performance”):

Among the bulk deconvolution methods, least-squares (OLS, nnls), support-vector (CIBERSORT) and robust regression approaches (RLR/FARDEEP) gave the best results across different datasets and pseudo-bulk cell pool sizes (median RMSE values < 0.05; Figure 3a, Supplementary Figures 10-12).

10) I understand the difficulty of using real-life data in such a benchmarking, however, there are datasets where certain cell types have some expected ranges (e.g., some type of immune cells should be absent in certain tissues). Can this be used to make a stronger point for the comparison of deconvolution methods in **real** datasets?

We thank the reviewer for raising this point (which was also raised as remark #1 from reviewer 3). We fortunately found, downloaded and processed the following raw RNA-seq (single-end) data of nine human (bulk) PBMCs samples from Finotello *et al.* (2019)¹⁰ for which cell type proportions were measured by flow cytometry (assumed “gold standard”); with T cells and Monocytes being the two most abundant cell types.

donor_ID	SRA_ID
donor_1	SRR6337113
donor_10	SRR6337120
donor_12	SRR6337121
donor_2	SRR6337114
donor_4	SRR6337115
donor_5	SRR6337116
donor_6	SRR6337117
donor_7	SRR6337118
donor_9	SRR6337119

We aligned the RNA-seq reads against the human genome (Homo sapiens; Ensembl v91; GRCh38) with STAR v2.6.0c and the output .bam files from the previous step were used as input for HTSeq v0.11.0, resulting in a final matrix with gene counts.

To answer this question we used the following data:

- dataset 1: bulk RNA-seq data of these nine PBMC samples for which flow cytometry based immune composition is known (acknowledging that the neutrophils (2.4 to 5%) were not present in dataset 2)
- dataset 2: scRNA-seq data from PBMCs (10x Genomics; see Table 1 from the main manuscript) (excluding the CD34+ cells as they were not in dataset 1).
- dataset 3: bulk RNA-seq for the following immune cell types: B cells, Monocytes, Myeloid DCs, NK and T cells (see comment #6 above for SRA accession IDs).

We set up two scenarios that allowed a fair deconvolution of the nine PBMC samples with 1) bulk deconvolution methods and 2) those able to use scRNA-seq data as reference (DWLS, deconvSeq and BisqueRNA. It was not possible to evaluate MuSiC and SCDC because the scRNA-seq data was obtained from only one individual):

	heterogeneous input	reference matrix	marker information	expected proportions
bulk deconvolution methodologies	9 PBMCs	bulk RNA-seq from cell types described in remark 6 excluding neutrophils	markers for Monocytes, B cells, NK, Dendritic cells and T cells obtained from 10x scRNA-seq data	flow cytometry (ignoring neutrophils)
deconvolution methods using scRNA-seq as reference	9 PBMCs	10x scRNA-seq data without CD34+	no prior information	flow cytometry (ignoring neutrophils)

Figure 7 – Deconvolution performance on nine human PBMC bulk samples using a) bulk deconvolution methods; (b) deconvolution methods using scRNA-seq as reference.

Regarding bulk deconvolution methods: robust regression methods (RLR, FARDEEP) and support vector regression (CIBERSORT) consistently showed the smallest RMSE and highest Pearson correlation values. Similarly, DWLS performed best among the deconvolution methods that use scRNA-seq data as input.

The new figure has also been included in the main manuscript (Figure 7) and this analysis and conclusion have been included in: two sections of the main manuscript entitled “Deconvolution performance using real bulk heterogeneous samples from human PBMCs” (Results section) and “Computational framework for the evaluation of deconvolution pipelines with real RNA-seq data” (Methods); one section in the supplementary material: “Processing poly(A) RNA-seq of nine human bulk PBMCs samples”.

11) The meaning and scale of RMSE as a measure of deconvolution quality is unclear. Can the authors provide some arguments why, for example, RMSE lower than 0.05 is a good deconvolution result as they claim.

We see how the statement “ [...] able to achieve very accurate cell type proportions, with median RMSE values lower than 0.05 ” could have been misleading. There is no “universally” good RMSE threshold, and it is also scale-dependent on the dependent variable of interest (in our case, cell type proportions ranging from 0 to 1). The goal is to always achieve an RMSE as small as possible. We have adapted the manuscript (in red) as follows:

In terms of performance, the five best bulk deconvolution methods (OLS, nnls, RLR, FARDEEP and CIBERSORT) and three best methods that use scRNA-seq data as reference (DWLS, MuSiC, SCDC) achieved median RMSE values lower than 0.05.

12) The Figure 2 does not seem informative or necessary to me. The timing of most of the methods and memory use seem to be reasonable and not limiting for practical studies. My suggestion is to move it to Supplementary. Also, I think it is better to report the computation time per sample.

We agree with the reviewer and have moved it to Supplementary material (Supplementary Figure 5; please see answer to point 4 from reviewer 1). Since all scRNA-seq datasets were used to generate the same amount of pseudo-bulk mixtures ($n = 1,000$), the conclusions from a comparison would remain the same as the global time. Since it has become part of the Supplementary material, we suggest that there is no additional value in including a new figure.

13) For the “all markers” strategy, it would be interesting to know if there were any overlap in genes as it was stated before that previous analyses speculated that the lack of specificity in markers explained some bad results for proportion estimations.

We thank the reviewer for making this very relevant point. For all markers across each dataset, we took a closer look at the fold-change distribution for both the cell type where they were initially found as marker (highest fold changes) and the fold-change differences among all other cell types. Using the threshold values used to select a gene as marker, we computed the percentage of those that could also be considered markers for a secondary cell type (values between parentheses in the boxplots below). For the five datasets included in the benchmark, 7 to 38% of the markers were not “specific” (exclusive) for only one cell type.

Supplementary Figure 2 – Fold change values for the different cell-type specific markers across the different datasets used in the manuscript. Left boxplots depict the fold change for the cell type where markers were originally found and right boxplots depict the fold changes for the remaining cell types present in the dataset.

This analysis and figure have been included in the Supplementary material as “Fold change evaluation of cell-type-specific markers used with bulk deconvolution methodologies” and Supplementary Figure 2, respectively. We have added the following paragraph inside “The set of markers used in bulk deconvolution methods impacts deconvolution results” section (Results):

For “all” markers across each dataset, we took a closer look at the fold-change distribution for both the cell type where they were initially found as marker (highest fold change) and the fold-change differences among all other cell types. Using the threshold values used to select a gene as marker, we computed the percentage of those that could also be considered markers for a secondary cell type (values between parentheses in the boxplots below). For the five datasets included in the benchmark, 7 to 38% of the markers were not “specific” (exclusive) for only one cell type (see Supplementary Figure 2).

14) In the abstract, the authors claim that 'the choice of the reference marker signatures is far more important than the method itself'. I think this can not be stated as a well-established fact. How many 'attempts' confirm this? The problem is that in many situations it is not easy to test all combinations of markers and methods, so I do not think this statement was 100% demonstrated.

That statement was actually not directly addressed in our manuscript but in another publication (Vallania *et al.*¹¹; see line 74 from the Introduction). Therefore, we have rephrased the first paragraph of the abstract as follows (changes in red):

Many computational methods to infer cell type proportions from bulk transcriptomics data have been developed. Previous attempts from other groups comparing these methods revealed that the choice of reference marker signatures is far more important than the method itself.

15) Of note, I did not want to access the private GitHub provided in the manuscript for anonymity reasons, therefore I made my evaluations based on the public repository https://github.com/favilaco/deconv_benchmark.

We apologize for this. The code was deployed using Ghent University's private Github account (https://github.ugent.be/favilaco/deconv_benchmark) and the public https://github.com/favilaco/deconv_benchmark did not contain the most recent README file. Both versions have been updated and now contain the same code/files.

16) Currently, the code accompanying the study is organized in one single R script. Since the code generating the synthetic data can be reused in other studies with different methods or strategies to generate the synthetic data, I suggest to reorganize the code into 1) synthetic data generator part, 2) various method caller part following unified interface, 3) proper benchmarking part.

1) We have updated the "Generator" function (inside "helper_functions.R") in our Github repository and have explained in the README that it can be used as a stand-alone function for users only interested in generating mixtures from scRNA-seq data. Furthermore, now it is possible to generate mixtures containing two - n cell types (where n is the maximum number of cell-types available in the scRNA-seq data used as input) with uniformly random proportions from 0.01 - 0.99 (1 to 99%) while simultaneously ensuring its sum to be one (100%). [Removed the cap of maximum 5 cell types and increasing proportions in steps of 0.05].

2) The function "Deconvolution" inside "helper_functions.R" can be easily enlarged by appending a new "else if" statement, enabling users to test additional methods if desired.

3) The usage of *helper_functions.R* and *Master_deconvolution.R* together combines the usage of a specific transformation, normalization, deconvolution method and, optionally, removal of a specific cell type from the reference matrix (= benchmarking part). The output already consisted in RMSE and Pearson correlation values (see README file in Github).

17) Test run for the Single cell example gave me RMSE=0.0821538, Pearson=0.934689 versus documented RMSE=0.06977075, Pearson=0.9527703. It is relatively minor but still 17% RMSE difference. The code was run on a Mac with all dependency packages installed and updated, using R version 3.6.1. Where might these differences come from? Can the OS/platform have an impact on the benchmarking result in general and can it alter the quality of some analyses? Has this code been tested on various platforms to check for reproducibility?

As outlined above, the code was deployed using Ghent University's private Github (https://github.ugent.be/favilaco/deconv_benchmark) and the public https://github.com/favilaco/deconv_benchmark did not contain the most recent README file. Both versions have been updated and contain the same code/files now.

We have fixed the RMSE and Pearson values displayed in README.md from https://github.com/favilaco/deconv_benchmark. Of note, these have been updated with the results obtained with the new "Generator" function (see answer to previous remark).

Reproducibility has been tested in these two different operative systems (OS):

R version 3.6.2 (2019-12-12)

Platform: x86_64-pc-linux-gnu (64-bit)

Running under: Linux Mint 19.3

R version 3.6.2 (2019-12-12)

Platform: x86_64-apple-darwin15.6.0 (64-bit)

Running under: macOS Sierra 10.12.6

The results were identical between Linux and macOS (see “sessionInfo_Linux.txt” and “sessionInfo_macOS.txt” in Github for the specific versions loaded in memory to perform such comparison).

Minor remarks:

18) Supplementary Figure 10 C,D are not readable. They can be zoomed in a PDF of course but I would suggest to make them printable.

We have re-ordered and enlarged the panel figures in order to make them more readable. (see Supplementary Figure 14).

19) In Figure 2 the panels are not described in their order in the figure legend. This makes it difficult to digest.

Figure 2 has been moved to Supplemental material (Supplementary Figure 5). The following (in red) has been added to its legend to clarify it:

RAM memory (bytes) (left hand side) and time (seconds) (right hand side) requirements for the different bulk deconvolution methodologies (top panel) and deconvolution methods using single-cell RNA-seq data as reference (bottom panel) across datasets with expression values in linear scale (boxplots depict all scaling/normalization strategies across all pseudo-bulk cell pool sizes).

20) In Line 178, I am confused by the terminology used. I discover that supervised methods assumes expression values for markers (weights) while the methods using sets of marker genes are called semi-supervised. So, MCPCounter would be a semi-supervised method (I do not agree with this). Note also that any unsupervised method also uses sets of marker genes for interpreting the meaning of factors/clusters.

We are sorry for that confusion here. It was Gaujoux and Seoighe¹² who implemented (and chose the name) of ssFrobenius and ssKL, where “ss” stands for semi-supervised. Nevertheless, the way the marker information is used differs from other methods (namely MCPCounter). Both ssFrobenius and ssKL are two non-negative matrix factorization approaches (unsupervised) that include an initial step where the initial matrix of heterogenous samples (in our case pseudo-bulk mixtures) is subset such that only rows of the marker genes are kept prior to generating two output matrices (matrix of cell type proportions and matrix with cell-type specific “expression” values) using an alternating least squares scheme (see also <https://web.cbio.uct.ac.za/~renaud/CRAN/web/CellMix/vignettes/SampleAnalysis.pdf>).

21) Figure 6c is difficult to read in print. Can it be improved?

We have re-ordered and enlarged the panel figures (now re-labelled as Figure 5) in order to make them more readable.

22) Lines 395-396 the statement about 'mean collapsing' is not clear to me.

We have re-written the paragraph as follows:

For the deconvolution methods that use scRNA-seq data as reference and for the marker selection step, the “training” subsets were used in their original single-cell form, whereas a mean gene expression collapsing step (= mean expression value across all cells of the same cell type) was required to generate the reference matrices used in the bulk deconvolution methods.

23) Lines 422-424 : considerations about possible pool sizes are not clear to me.

We apologize for the confusion and have re-written the paragraph as follows:

The minimum number of cells used to create the pseudo-bulk mixtures (pool size) for each of the five datasets was 100 and the maximum possible number was determined by the second most abundant cell type (rounded down to the closest hundred, to avoid non-integer numbers of cells), resulting in $n = 100$, 700 and 1200 for Baron; $n = 100$, 300 and 400 for PBMCs; $n = 100$ and 200 for GSE81547; $n = 100$ and 200 for the kidney dataset and $n = 100$ for E-MTAB-5061.

Reviewer #3 (Remarks to the Author):

In this manuscript, Cobos and colleagues evaluate deconvolution methods (and associated preprocessing strategies) using simulations involving pseudo-bulk single-cell RNA-seq data. They generate some measures of performance for each method from datasets and provide some recommendations for obtaining best results. The manuscript is clear and addresses an interesting problem; however, there are some deficiencies with the evaluation framework that make it challenging to extrapolate the results to real-life application of deconvolution methods. I have listed my concerns in more detail below.

1) I understand that the authors' evaluation is based purely on pseudo-bulk single-cell RNA-seq profiles. However, the methods under discussion are intended to operate on actual bulk RNA-seq profiles. There is a subtle but important distinction here, as bulk RNA-seq remains quite different from the library preparation technology used in single-cell experiments. I list a few such differences below:

- Many bulk protocols use random primers compared to the poly-A approach that dominates single-cell protocols, driving differences in capture efficiency across genes.
- Bulk protocols tend to have much fewer rounds of PCR, reducing amplification biases.
- Bulk protocols can also afford to undergo RNA purification and other clean-up steps (e.g., ribosomal depletion).
- Bulk protocols preserve differences in RNA content between multiple cell types in a mixed population, whereas such differences are difficult to recover in single-cell data without spike-ins.
- Bulk data is typically full-length without UMIs, so using pseudo-bulk UMI counts is especially unrealistic.

All of these factors can skew the observed expression profile such that the bulk counts are not simply the sum of single-cell counts.

Why do I mention this? Well, testing methods on pseudo-bulk profiles gives an overly optimistic impression of the performance of each method. The simulated mixtures have exactly the same properties as the reference profiles, which seems divorced from real applications of deconvolution methods. The current evaluation does not capture the robustness of each method to technical differences between protocols such as, e.g., changes in capture efficiency for key marker genes. **It would seem more meaningful to construct the mixtures using bulk RNA-seq from pure populations (e.g., after FACS sorting).** There are a number of datasets where such data are available; ImmGen or the Database of Immune Cell Expression come to mind. The evaluation would then focus on the accuracy of using single-cell reference profiles to deconvolve those bulk-derived mixtures in an accurate yet robust manner.

We thank the reviewer for raising this point (which was also raised as remark #10 from reviewer 2). We downloaded and processed raw poly(A) RNA-seq (single-end) data of nine human (bulk) PBMCs samples from Finotello *et al.*¹⁰ for which cell type proportions were measured by flow cytometry (assumed “gold standard”; see Suppl. Table 2 and “Processing poly(A) RNA-seq of nine human bulk PBMCs samples” in Supplementary Methods). Furthermore, we used scRNA-seq data from PBMCs (10x Genomics; see Table 1) and bulk RNA-seq for B cells, monocytes, myeloid dendritic cells, natural killer and T cells (see Supplementary Table 1).

Importantly, we acknowledge two limitations in this set-up: i) the nine PBMCs include a measurement for neutrophils (2.45% to 5.05%) while such cell type was not present in the 10x scRNA-seq data; ii) The 10x scRNA-seq data contained CD34+ cells whereas the nine PBMCs did not include information for such cell type.

Therefore, to establish an unbiased assessment for both bulk deconvolution methods and those that use scRNA-seq data as reference, we excluded CD34+ cells from the 10x scRNA-seq data and did not use the bulk RNA-seq data for neutrophils in the reference matrix (see Table 2). Of note, flow cytometry proportions for T cells were computed as the sum of proportions of three different sub-populations (T_{reg} s, CD8+ and CD4+).

Scenarios used to assess the performance of the different deconvolution methods using real mixtures (bulk RNA-seq from nine PBMCs samples):

	heterogeneous input	reference matrix	marker information	expected proportions
bulk deconvolution methodologies	9 human bulk PBMCs	bulk RNA-seq of monocytes, B cells, natural killer, dendritic cells and T cells (SRA accession from Suppl. Table 1)	markers for monocytes, B cells, natural killer, dendritic cells and T cells obtained from 10x scRNA-seq data	flow cytometry measurements
deconvolution methods using scRNA-seq as reference	9 human bulk PBMCs	10x scRNA-seq data excluding CD34+ cells	no prior information	flow cytometry measurements

To note, it was not possible to evaluate MuSiC and SCDC because the 10x scRNA-seq data used as reference came from only one individual. Hence, only DWLS, deconvSeq, and BisqueRNA were tested. See “Computational set-up for the evaluation of deconvolution pipelines with real RNA-seq data” (Methods) and Table 1 for more details.

Figure 7 – Deconvolution performance on nine human PBMC bulk samples using a) bulk deconvolution methods; (b) deconvolution methods using scRNA-seq as reference.

Regarding bulk deconvolution methods: robust regression methods (RLR, FARDEEP) and support vector regression (CIBERSORT) consistently showed the smallest RMSE and highest Pearson correlation values. Similarly, DWLS performed best among the deconvolution methods that use scRNA-seq data as input.

The new figure has also been included in the main manuscript (Figure 7) and this analysis and conclusion have been included in: two sections of the main manuscript entitled “Deconvolution performance using real bulk heterogeneous samples from human PBMCs” (Results section) and “Computational framework for the evaluation of deconvolution pipelines with real RNA-seq data” (Methods); one section in the supplementary material: “Processing poly(A) RNA-seq of nine human bulk PBMCs samples”.

2) The authors focus on the RMSE between the predicted and true cell type proportions to determine method performance. This is hard to interpret; I would instead like to know the error relative to the magnitude of the proportion. For example, a method with an RMSE of 0.05 for a cell type might not be so bad if that type's true proportion is 0.5, but it would be much less appealing if the true proportion is 0.01.

We thank the referee for making this point. Here, RMSE is not computed for one point. Instead, it is calculated for all cell types composing one mixture (or for all mixtures if we consider a combination of transformation, normalization and methodology). Therefore, we feel that it is difficult to imagine a “RMSE for one cell type with a known proportion of 0.5”.

However, to investigate the performance at the proportion level, we defined a measure of relative error as follows:

$$relative\ error = \frac{abs(computed\ proportion - expected\ proportion)}{expected\ proportion}$$

This measurement (due to the denominator) takes into account “the magnitude” of the error: an over/under-estimation of 0.05 is barely relevant if the expected proportion was 0.9 but very important if the expected was 0.1. From this new analysis we showed the results for GSE81547 using nnls, CIBERSORT (two bulk deconvolution methods); DWLS and MuSiC (two deconvolution methods that use scRNA-seq data as input) and pool sizes of 100 and 200 cells.

Supplementary Figure 20 – GSE81547 dataset: regardless of the cell type (each row), pseudo-bulk cell size ($n = 100, 200$) and bulk deconvolution method (nnls, CIBERSORT) being investigated, small proportions are always more difficult to reconstruct (higher relative errors) than high proportion values.

Supplementary Figure 21 – GSE81547 dataset: regardless of the cell type (each row), pseudo-bulk cell size ($n = 100, 200$) and deconvolution method that use scRNA-seq data as reference (DWLS, MuSiC) being investigated, small proportions are always more difficult to reconstruct (higher relative errors) than high proportion values.

Regardless of the method, cell type and pool size evaluated, small proportions are always more difficult to reconstruct (higher relative errors) than high proportion values. We repeated the analysis for the E-MTAB-5061 dataset and the conclusions were identical. We have included the figures for GSE81547 and E-MTAB-5061 datasets in the Supplementary material (Supplementary Figures 20-23) and the following has been added to the main manuscript (Results section; changes in red):

When considering the estimation error relative to the magnitude of the expected cell type proportions, smaller proportions consistently showed higher relative errors (see Supplementary Figures 20-23). Of note, quantile normalization [...]

3) On a related note, I would say that the absolute proportion holds little biological meaning by itself. Much more interesting is the difference in proportions between conditions, e.g., does a particular cell type increase in abundance before and after drug treatment? To this end, a useful evaluation would look at whether relative differences in the true cell type proportions are preserved in the deconvolution output.

We agree with the reviewer, changes in cell type proportions upon a drug treatment or a viral infection are indeed relevant (e.g. Altboum *et al.*, 2014¹³ showed differences in immune cell type composition during the course of influenza in mice lung). Nevertheless, cell type composition of the tumor micro-environment on its own (e.g. at baseline level, before any treatment has started) is also very relevant in the field of immune-oncology (e.g. patients with high levels of tumor-infiltrating lymphocytes responded better to immune checkpoint inhibitors (immunotherapy)¹⁴). In principle, the performance of the computational deconvolution should be independent of the experimental set up where it is applied. Nevertheless, we acknowledge that our benchmark did not specifically evaluate the former scenario.

We have added this insight/shortcoming to the Discussion:

Both cell type proportions on their own (e.g. at baseline level, before any treatment has started) and changes in cell type composition upon drug treatment or a viral infection are relevant and can be assessed through computational deconvolution. For instance, patients with high levels of tumor-infiltrating lymphocytes have been found to respond better to immune checkpoint inhibitors (immunotherapy)¹⁴ and changes in diverse immune cell types were found in mice lungs during the course of influenza infection¹³. In principle, the performance of a computational deconvolution framework should be independent of the experimental set up where it is applied. However, we acknowledge that the data included in our benchmark did not directly evaluate the latter scenario.

4) It was not clear to me what proportions were used to construct the mixtures. I would hope that the authors created mixtures with increasingly rare populations to examine the performance of each method for recovering correct values when the true proportions of each cell type span a wide range of values.

We are sorry for the confusion, the answer to that question can be found in the section “Generation of thousands of artificial pseudo-bulk mixtures” (Methods). We have also added the information about the new dataset included (kidney), following comments #8 and #9 from reviewer 2 (changes in red):

For the human pancreas and PBMC datasets, each (feasible) pseudo-bulk mixture was created by randomly (uniformly) selecting the number of cell types to be present (between 2 and 5) and their identities, followed by choosing the cell type proportion assigned to each cell type (enforcing a sum-to-one constraint) among all possible proportions between 0.05 and 1, in increasing intervals of 0.05. For the kidney data, the number of cell types present was randomly (uniformly) selected between 2 and 8 (eight being the maximum number of cell types possible), followed by selecting the cell type proportion assigned to each cell type (enforcing a sum-to-one constraint) among all possible proportions between 0.01 and 0.99. Finally, once the number of cells to be picked up from specific cell types was determined, the cells were randomly selected without replacement (= a given cell can only be present once in a mixture).

5) The authors mention that the methods operate best on non-transformed data. This is nice to know but is this not obvious? The bulk mixture is constructed (in silico or in the lab) by pooling expression profiles on the original scale, so it seems clear that the best approach would be to perform deconvolution on the same scale. It would be helpful to have some elaboration on why this is not a trivial result.

The use of the expression data in its linear form instead of the log-transformed version is an important difference with respect to classical differential gene expression analyses, where statistical tests assume underlying normal distributions therefore requiring log transformation. We do not claim this is a new finding, but we thought it was worth checking and our results agree with Zhong and Liu².

Please, see also the answer to remark #6 from reviewer 2 (real bulk transcriptomes can be approximated as linear mixtures of the constituent cell types).

We have added the following to the Discussion (red):

The logarithmic transformation is routinely included as a part of the pre-processing of omics data in the context of differential gene expression analysis^{15,16}, but Zhong and Liu² showed that it led to worse results than performing computational deconvolution in the linear (un-transformed) scale. The use of the expression data in its linear form is an important difference with respect to classical differential gene expression analyses, where statistical tests assume underlying normal distributions (typically achieved by the logarithmic transformation¹⁷).

6) It would be necessary to show that the differences in the methods have some kind of impact on real data analyses. I do not expect the authors to evaluate the methods on real data or to generate a gold-standard mixture experiment; I just want to see that any differences in behavior observed in the simulations are reflected in real data. For example, if one method struggles to detect rare populations in simulations, we should also see a dearth of small proportions when applied on real data.

Regarding the differences across methods with real data analysis, please see our answer to your remark #1. For a detailed overview of the error across different expected cell type proportions, please see our answer to your remark #2.

References

1. Sturm, G. *et al.* Comprehensive evaluation of transcriptome-based cell-type quantification methods for immuno-oncology. *Bioinformatics* **35**, i436–i445 (2019).
2. Zhong, Y. & Liu, Z. Gene expression deconvolution in linear space. *Nat. Methods* **9**, 8–9 (2012).
3. Vieth, B., Parekh, S., Ziegenhain, C., Enard, W. & Hellmann, I. A systematic evaluation of single cell RNA-seq analysis pipelines. *Nat. Commun.* **10**, 1–11 (2019).
4. Zyprych-Walczak, J. *et al.* The Impact of Normalization Methods on RNA-Seq Data Analysis. *BioMed Research International* <https://www.hindawi.com/journals/bmri/2015/621690/> (2015) doi:<https://doi.org/10.1155/2015/621690>.
5. Lytal, N., Ran, D. & An, L. Normalization Methods on Single-Cell RNA-seq Data: An Empirical Survey. *Front. Genet.* **11**, (2020).
6. Abdelaal, T. *et al.* A comparison of automatic cell identification methods for single-cell RNA sequencing data. *Genome Biol.* **20**, 194 (2019).
7. Avila Cobos, F., Vandesompele, J., Mestdagh, P. & De Preter, K. Computational deconvolution of transcriptomics data from mixed cell populations. *Bioinformatics* **34**, 1969–1979 (2018).
8. Scialdone, A. *et al.* Computational assignment of cell-cycle stage from single-cell transcriptome data. *Methods* **85**, 54–61 (2015).
9. Han, X. *et al.* Construction of a human cell landscape at single-cell level. *Nature* 1–7 (2020) doi:10.1038/s41586-020-2157-4.
10. Finotello, F. *et al.* Molecular and pharmacological modulators of the tumor immune contexture revealed by deconvolution of RNA-seq data. *Genome Med.* **11**, 34 (2019).
11. Vallania, F. *et al.* Leveraging heterogeneity across multiple datasets increases cell-mixture deconvolution accuracy and reduces biological and technical biases. *Nat. Commun.* **9**, 4735 (2018).
12. Gaujoux, R. & Seoighe, C. Semi-supervised Nonnegative Matrix Factorization for gene expression deconvolution: A case study. *Infect. Genet. Evol.* **12**, 913–921 (2012).
13. Altboum, Z. *et al.* Digital cell quantification identifies global immune cell dynamics during influenza infection. *Mol. Syst. Biol.* **10**, 720 (2014).
14. Darvin, P., Toor, S. M., Nair, V. S. & Elkord, E. Immune checkpoint inhibitors: recent progress and potential biomarkers. *Exp. Mol. Med.* **50**, 1–11 (2018).
15. Gene Expression Studies Using Affymetrix Microarrays. *CRC Press* <https://www.crcpress.com/Gene-Expression-Studies-Using-Affymetrix-Microarrays/Gohlmann-Talloe/p/book/9781138112315>.
16. Zwiener, I., Frisch, B. & Binder, H. Transforming RNA-Seq Data to Improve the Performance of Prognostic Gene Signatures. *PLOS ONE* **9**, e85150 (2014).
17. Hoyle, D. C., Rattray, M., Jupp, R. & Brass, A. Making sense of microarray data distributions. *Bioinformatics* **18**, 576–584 (2002).

Reviewer #1 (Remarks to the Author):

The authors have made relatively limited changes to the manuscript in response to my original review. They disagree on whether the manuscript presents novel findings or mostly a review, providing comprehensive evaluation of existing methods.

The authors agree with the second point raised ("It does not really make sense to compare performance of the semi-supervised and supervised methods –the scenarios in which they would be used and the input data are different..."). This should at least have been acknowledged in the revised manuscript (the semi-supervised methods can be applied to data to which the supervised methods cannot as they do not require reference expression levels).

The revised manuscript does not go far enough to address the potential issue arising from data from the same samples occurring in the training and test sets. It appears from the authors' response to this point that even when data from multiple individuals were available, cells from the same individuals occurred in the training and test data. The authors state that the "observed meaningful differences between cell types rather than by individual". While it's true, not surprising and well illustrated in Fig. S24, that the differences in expression between cell types tends to be far greater than the differences in expression between the same cell type in different individuals, this is not sufficient to be confident that including cells from the same sample in the training and test data may have affected estimates of method performance. Some of the datasets had sufficient individuals for the authors to have separated the samples fully between training and test datasets. The authors should estimate the performance of methods when they are trained on one group of individuals and applied to another, treating each individual in the test data as a single data point, rather than effectively throwing away inter-individual and technical variation by mixing cells from the same samples in both the training and test data. This would be equivalent to the real world setting, in which the reference samples and separate to the samples on which the deconvolution methods are applied. If they find that this does not affect performance using the samples where they have enough individuals to test in this way, that would help to allay concerns over mixing of samples between training and test data in the case where there are insufficient samples to apply the full separation. In my view this is an important issue that needs to be addressed.

Reviewer #2 (Remarks to the Author):

Comparing the the initial version of the manuscript, I see that the authors have carefully addressed all my remarks. I am satisfied with the new version which I believe provides a stronger message.

Reviewer #3 (Remarks to the Author):

The authors have addressed all of my concerns.

Point-by-point response to the referees' comments

We would like to thank the reviewers for thoroughly reading our manuscript and their constructive suggestions, as we believe they have improved our manuscript. We have adjusted our manuscript according to the reviewers' comments and the present document provides detailed answers (blue) to the remaining comments of reviewer 1 (in black).

We have also:

- Unified the colour scale for Pearson correlation palette values (always from -1 to +1 instead of dataset-dependent [min.Pearson, max.Pearson] ranges) across all figures both in main and supplemental.
- Changed: "none" by "linear"; "vst" by "VST"; "PBMCs2" to "PBMCs" and "RNBR" by "SCTransform".
- Ensured that the data transformations have consistent colours across all figures (gray for linear, orange for logarithmic, blue for square root and green for VST).
- We have added the results for the kidney.HCL dataset in those figures where they were missing (Supplementary Figures 10-11).
- Supplementary Figure 11 became difficult to read after including a fifth dataset (kidney.HCL), so it has been split into Suppl. Figure 11a and 11b (over two pages) to increase readability.

Reviewer #1 (Remarks to the Author):

The authors agree with the second point raised ("It does not really make sense to compare performance of the semi-supervised and supervised methods –the scenarios in which they would be used and the input data are different.."). This should at least have been acknowledged in the revised manuscript (the semi-supervised methods can be applied to data to which the supervised methods cannot as they do not require reference expression levels).

We agree with the reviewer. We have added the following paragraph in the Discussion:

In general, the use of all data at hand (i.e. in supervised strategies) leads to better results than unsupervised or semi-supervised approaches. However, in other contexts different from computational deconvolution (e.g. automatic cell identification¹), it has been shown that incorporating prior knowledge into the models does not improve the performance. Furthermore, there are situations where cell-type specific expression profiles are not readily available and supervised methodologies cannot be used. For these reasons we included ssFrobenius and ssKL in our benchmarking, two semi-supervised non-negative matrix factorization methods to perform bulk gene expression deconvolution. They led to higher RMSE and lower Pearson correlation values than most supervised methodologies (except DCQ and dtangle; Figure 2; Supplementary Figure 6), highlighting the positive impact of incorporating prior knowledge (in the form of cell-type specific expression profiles) in the field of computational deconvolution. In any case, results from supervised and semi-supervised methodologies should be interpreted separately.

The revised manuscript does not go far enough to address the potential issue arising from data from the same samples occurring in the training and test sets. It appears from the authors' response to this point that even when data from multiple individuals were available, cells from the same individuals occurred in the training and test data. The authors state that the "observed meaningful differences between cell types rather than by individual". While it's true, not surprising and well illustrated in Fig. S24, that the differences in expression between cell types tends to be far greater than the differences in expression between the same cell type in different individuals, this is not sufficient to be confident that including cells from the same sample in the training and test data may have affected estimates of method performance. Some of the datasets had sufficient individuals for the authors to have separated the samples fully between training and test datasets. The authors should estimate the performance of methods when they are trained on one group of individuals and applied to another, treating each individual in the test data as a single data point, rather than effectively throwing away inter-individual and technical variation by mixing cells from the same samples in both the training and test data. This would be equivalent to the real world setting, in which the reference samples and separate to the samples on which the deconvolution methods are applied. If they find that this does not affect performance using the samples where they have enough individuals to test in this way, that would help to allay concerns over mixing of samples between training and test data in the case where there are insufficient samples to apply the full separation. In my view this is an important issue that needs to be addressed.

We thank the reviewer for raising this relevant point, which we had overlooked in previous versions of the manuscript. While our approach described in the section "Deconvolution performance using real bulk heterogeneous samples from human PBMCs" (added after the first peer review) partially addressed this issue (because the 9 heterogeneous samples came from one source whereas the scRNA-seq and markers came from another; see Table 2), we have now performed additional analyses to address this question.

Of the five scRNA-seq datasets included in the manuscript, the PBMC data consisted of a single individual, and the kidney.HCL data (with $n = 2$) would lead to a split of 1 individual for training and 1 for testing. Since MuSiC and SCDC methods cannot be used in these situations (they require multiple samples ($n > 1$) for the reference matrix, which was built with the training split), these datasets were discarded.

We proceeded with the three datasets that contained data on more than 2 samples, and split each one into 2 new groups containing half of the samples each (keeping age and gender balance when possible), as described below. Samples in "group 1" were considered as the "training" split (to build the reference matrix and perform marker selection) whereas those in group 2 were considered the "testing" split (to make 1000 pseudo-bulk mixtures). We retained only those cell types with at least 50 cells in both groups.

Baron dataset:

- Group1 = "human1" (gender: male, age: 17) + "human2" (female, 51)
- Group2 = "human3" (male, 38) + "human4" (female, 59)
- Number of cells per cell type and group:

	Group1	Group2
acinar	63	506
activated_stellate	122	138
alpha	904	1398
beta	1202	1252
delta	337	259
ductal	392	574
endothelial	153	99
gamma	155	99
quiescent_stellate	114	57

GSE81547 dataset:

- Group1 = "1yr_male" + "21yr_male" + "54yr_male" + "38yr_female"
- Group2 = "5yr_male" + "6yr_male" + "22yr_male" + "44yr_female"
- Number of cells per cell type and group:

	Group1	Group2
acinar	240	171
alpha	592	406
beta	157	191
ductal	228	161

E-MTAB-5061 dataset:

- Group1 = H3 (female, 48) + H4 (male, 22) + H5 (male, 27)
- Group2 = H1 (male, 43) + H6 (male, 23) + H2 (male, 25)
- Number of cells per cell type and group:

	Group1	Group2
alpha_cell	206	237
beta_cell	76	95

Since E-MTAB-5061 would lead to mixtures containing only two cell types and it shares the same sequencing method as GSE81547 (Smart-Seq2; see Table 1), we have re-run all deconvolution frameworks (both bulk and single-cell) with the new data splits for the Baron and GSE81547 datasets (included as Supplementary Figures 25 and 26, respectively):

Supplementary Figure 25 – RMSE and Pearson correlation values between the expected (known) proportions in 1000 pseudo-bulk tissue mixtures in linear scale (pool size = 100 cells per mixture) and the output proportions from the different bulk deconvolution methods for Baron and GSE81547 datasets (top and bottom panel, respectively). “Same individuals” represent scenarios where both training and test sets contained cells from all individuals whereas “different individuals” represent results where training and test splits were made of cells from independent individuals.

Supplementary Figure 26 – RMSE and Pearson correlation values between the expected (known) proportions in 1000 pseudo-bulk tissue mixtures in linear scale (pool size = 100 cells per mixture) and the output proportions from the different deconvolution methods that use scRNA-seq data as input (for Baron and GSE81547 datasets; top and bottom panel, respectively). “Same indiv.” represent scenarios where both training and test sets contained cells from all individuals whereas “different indiv.” represent results where training and test splits were made of cells from independent individuals.

The cases where cells from a given individual were used only in one split (training or test) led to slightly higher RMSE and lower Pearson correlation values compared to those where cells from one individual were present in both splits, but the original conclusions still hold true.

We have added the following information to the Discussion section (in red):

Given the limited number of cells available per dataset and the scarcity of publicly available datasets with similar health status, sequencing platform and library preparation protocol to validate our results, some cells were used in more than one mixture and each dataset was split into training and testing (50%:50%), meaning that cells from one individual were present both in training and test sets but a given cell was only present in one split. Nevertheless, while the different datasets (except PBMCs) contain cells from more than one individual (= inherent inter-sample variability), we observed meaningful differences between cell types rather than by individual (Supplementary Figure 24). Additionally, we generated scenarios where cells from a given individual were used only in one split (training or test) by assigning half of the samples to each split prior to selecting the cells based on the cell type. These led to slightly higher RMSE and lower Pearson correlation values compared to those where cells from one individual were present in both splits, but the same conclusions hold true in both analyses (Supplementary Figures 25-26).

References

1. Abdelaal, T. *et al.* A comparison of automatic cell identification methods for single-cell RNA sequencing data. *Genome Biol.* **20**, 194 (2019).

REVIEWERS' COMMENTS:

Reviewer #1 (Remarks to the Author):

The authors have carried out additional work that addresses the main concerns I had after the previous review.